# Joint Feature and Differentiable $k$-NN Graph Learning using Dirichlet Energy

**Lei Xu**
School of Computer Science &
School of Artificial Intelligence,
OPtics and ElectroNics (iOPEN)
Northwestern Polytechnical University
Xi'an 710072, P.R. China
`solerxl1998@gmail.com`

**Lei Chen**
School of Computer Science
Nanjing University of Posts and Telecommunications
Nanjing 210003, P.R. China
`chenlei@njupt.edu.cn`

**Rong Wang**
School of Artificial Intelligence,
OPtics and ElectroNics (iOPEN)
Northwestern Polytechnical University
Xi'an 710072, P.R. China
`wangrong07@tsinghua.org.cn`

**Feiping Nie**[*]
School of Artificial Intelligence,
OPtics and ElectroNics (iOPEN) &
School of Computer Science
Northwestern Polytechnical University
Xi'an 710072, P.R. China
`feipingnie@gmail.com`

**Xuelong Li**
School of Artificial Intelligence,
OPtics and ElectroNics (iOPEN)
Northwestern Polytechnical University
Xi'an 710072, P.R. China
`li@nwpu.edu.cn`

## Abstract

Feature selection (FS) plays an important role in machine learning, which extracts important features and accelerates the learning process. In this paper, we propose a deep FS method that simultaneously conducts feature selection and differentiable $k$-NN graph learning based on the Dirichlet Energy. The Dirichlet Energy identifies important features by measuring their smoothness on the graph structure, and facilitates the learning of a new graph that reflects the inherent structure in new feature subspace. We employ Optimal Transport theory to address the non-differentiability issue of learning $k$-NN graphs in neural networks, which theoretically makes our method applicable to other graph neural networks for dynamic graph learning. Furthermore, the proposed framework is interpretable, since all modules are designed algorithmically. We validate the effectiveness of our model with extensive experiments on both synthetic and real-world datasets.

## 1 Introduction

Feature selection (FS) is a critical technique in machine learning that identifies informative features within the original high-dimensional data. By removing irrelevant features, FS speeds up the learning process and enhances computational efficiency. In many real-world applications such as

---

[*]Corresponding author.

37th Conference on Neural Information Processing Systems (NeurIPS 2023).

image processing, bioinformatics, and text mining [1–3], FS techniques are widely used to identify important features, thereby providing some explanations about the results and boosting the learning performance [4–7].

While numerous FS methods have been proposed in both supervised and unsupervised settings, as several studies highlighted [8, 9], the nature of FS is more unsupervised due to the unavailability of task-specific labels in advance. The selected features should be versatile to arbitrary downstream tasks, which motivates us to focus on the unsupervised FS in this study. Related works have recently resorted to neural networks to exploit the nonlinear information within feature space. For example, AEFS [9] uses the group Lasso to regularize the parameters in the first layer of the autoencoder, so as to reconstruct original features based on the restricted use of original features. Another well-known method is CAE [8], which selects features by learning a concrete distribution over the input features. However, most unsupervised deep methods rely on the reconstruction performance to select useful features. On the one hand, if there exists noise in the dataset, the reconstruction performance will be terrible even if useful features are selected, since the noise cannot be reconstructed with these informative features (see our reconstruction experiments on Madelon in Section 4.3). On the other hand, it is difficult to explain why selected features reconstruct the original data well. These issues prompt us to seek a new target for unsupervised deep FS.

As the saying goes, "*birds of a feather flock together*", the homophily principle [10] suggests that similar samples tend to be connected in a natural graph structure within real-world data. This graph structure is useful for describing the intrinsic structure of the feature space, and is commonly used in machine learning studies [11, 12]. Building upon this graph structure, He et al. [13] introduce the Dirichlet Energy, which they call "locality preserving power", as a powerful tool for unsupervised FS that is able to identify informative features reflecting the intrinsic structure of the feature space.

In many practical applications, the graph structure is not naturally defined and needs to be constructed manually based on the input features using some similarity measurements. The quality of features affects the quality of the constructed graph. As highlighted in [14], the useless features increase the amount of unstructured information, which hinders the exploration of inherent manifold structure within data points and deteriorates the quality of constructed graph. Therefore, the reference [14] proposes the UDFS method to discard such nuisance features and constructs a $k$-nearest-neighbor (NN) graph on the selected features using the heat kernel. Despite the good performance of UDFS, constructing graphs using the heat kernel may not reflect the intrinsic structure of the feature space. Besides, the sorting algorithms in learning the $k$-NN graph in UDFS is non-differentiable in neural networks, which restricts its application in downstream networks.

In this paper: (1) We propose a deep unsupervised FS network that performs simultaneous feature selection and graph learning by minimizing the Dirichlet Energy, thereby revealing and harnessing the intrinsic structure in the dataset. (2) Within the network, a Unique Feature Selector (UFS) is devised to approximate discrete and distinct feature selection using the Gumbel Softmax technique combined with decomposition algorithms. (3) Moreover, a Differentiable Graph Learner (DGL) is devised based on the Optimal Transport theory, which is capable of obtaining a differentiable $k$-NN graph that more accurately reflects the intrinsic structure of the data than traditional graph constructing methods. Due to the differentiability, DGL is also theoretically capable of serving as a learnable graph module for other graph-based networks. (4) The entire framework is developed algorithmically. Unlike most deep learning networks with complex components that are tough to decipher, each core module in our framework has an algorithmic and physically interpretable design, which greatly facilitates observing and understanding the network's internal operations during the learning process. (5) Experimental results on both synthetic datasets and real-world datasets demonstrate the effectiveness of our method.

**Notations.** For an arbitrary matrix $M \in \mathcal{R}^{a \times b}$, $m^i$, $m_i$, and $m_{i,j}$ denote the $i$-th row, the $i$-th column, and the $(i, j)$-th entry of $M$, respectively. Given a vector $m \in \mathcal{R}^b$, its $\ell_2$-norm is defined as $\|m\|_2 = \sqrt{\sum_{i=1}^{b} m_i^2}$. Based on this, the Frobenius norm of $M$ is defined as $\|M\|_F = \sqrt{\sum_{i=1}^{a} \|m^i\|_2^2}$. When $a = b$, the trace of $M$ is defined as $\text{tr}(M) = \sum_{i=1}^{a} m_{i,i}$. Given two matrices $M, N \in \mathcal{R}^{a \times b}$, we define their inner product as $\langle M, N \rangle = \sum_{i=1}^{a} \sum_{j=1}^{b} m_{i,j} n_{i,j}$. $\mathbf{1}_b$ denotes a $b$-dimensional column vector with all entries being 1, and $I_b$ denotes a $b$-order identity matrix. $\text{Bool}(cond)$ is a boolean operator that equals 1 if $cond$ is true, otherwise it equals 0. Moreover, given a vector $m \in \mathcal{R}^b$, we define its sorting permutation in ascending order as $\sigma \in \mathcal{R}^b$, namely, $m_{\sigma_1} \leq m_{\sigma_2} \leq \cdots \leq m_{\sigma_b}$.

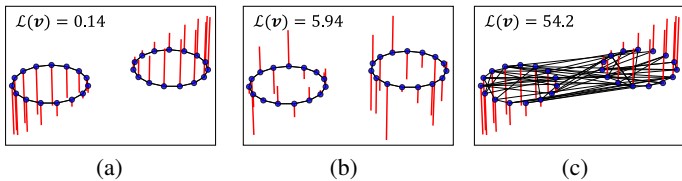

Figure 1: Illustration of the Dirichlet Energy on various graph structures and graph signals. Blue points, black edges, and red bars represent nodes, connections, and signal values on nodes, respectively. Upside bars represent positive values, and downside bars represent negative values.

## 2   Dirichlet Energy

Let $\boldsymbol{X} \in \mathcal{R}^{n \times d}$ be the data matrix with the $n$ samples and $d$-dimensional features. In this paper, we assume that the features have zero means and normalized variances,[2] namely, $\mathbf{1}_n^\top \boldsymbol{x}_i = 0$ and $\boldsymbol{x}_i^\top \boldsymbol{x}_i = 1$ for $i \in \{1, \ldots, d\}$. According to the homophily assumption [10], we assume that data $\boldsymbol{X}$ forms an inherent graph structure $\mathcal{G}$ with nodes standing for samples and edges standing for their correlations. In $\mathcal{G}$, similar samples are more likely to connect to each other than dissimilar ones. Graph $\mathcal{G}$ can be represented with a similarity matrix $\boldsymbol{S} \in \mathcal{R}_+^{n \times n}$, where $s_{i,j}$ denotes the similarity between $\boldsymbol{x}^i$ and $\boldsymbol{x}^j$. If $s_{i,j} = 0$, it means that there is no connection between $\boldsymbol{x}^i$ and $\boldsymbol{x}^j$ in $\mathcal{G}$, which is common in $k$-NN graphs since we only consider the local structure of the data space. Given an adjacency matrix $\boldsymbol{S}$,[3] we define the *Laplacian matrix* [15] as $\boldsymbol{L}_S = \boldsymbol{D} - \boldsymbol{S}$, where $\boldsymbol{D}$ is a diagonal matrix whose diagonal entries represent the degrees of data points, namely, $d_{i,i} = \sum_{j=1}^n s_{i,j}$.

Based on the Laplacian matrix $\boldsymbol{L}_S$, we introduce the *Dirichlet Energy* [16] as a powerful tool to identify important features. Specifically, given the Laplacian matrix $\boldsymbol{L}_S$, the Dirichlet Energy of a graph signal $\boldsymbol{v}$ is defined as

$$\mathcal{L}_{dir}(\boldsymbol{v}) = \frac{1}{2} \sum_{i=1}^n \sum_{j=1}^n s_{i,j}(v_i - v_j)^2 = \boldsymbol{v}^\top \boldsymbol{L}_S \boldsymbol{v}. \tag{1}$$

In graph theory, each dimensional feature $\boldsymbol{x}_i \in \mathcal{R}^n$ can be seen as a graph signal on $\mathcal{G}$. The Dirichlet Energy in Eq. (1) provides a measure of the local smoothness [11] of each feature on graph $\mathcal{G}$, which is small when the nodes that are close to each other on $\mathcal{G}$ have similar feature values. Hence, the Dirichlet Energy can be used to identify informative features by evaluating the consistency of the distribution of feature values with the inherent data structure. To demonstrate this, we provide an example in Fig. 1, where we generate a 2-NN graph $\mathcal{G}$ including two bubbles, and compose the data $\boldsymbol{X}$ using the two-dimensional coordinates of the graph nodes. Then we set the graph signal $\boldsymbol{v}$ as the first coordinate $\boldsymbol{x}_1$ and visualize it on $\mathcal{G}$ in Fig. 1(a). In Fig. 1(b) we change $\boldsymbol{v}$ to a random noise vector. While in Fig. 1(c), we change the graph structure to a random 2-NN graph. We compute the Laplacian matrix $\boldsymbol{L}_S$ of each figure and present the corresponding Dirichlet Energy in the figures. We can see that Fig. 1(a) achieves the best smoothness, whereas both Fig. 1(b) and Fig. 1(c) have poor smoothness due to a mismatch between the graph signal and the graph structure.

Based on the Dirichlet Energy, a well-known FS method called Laplacian Score (LS) [4] is proposed in [13]. However, the Laplacian matrix in LS is precomputed and fixed. If $\boldsymbol{X}$ contains too many irrelevant features, the quality of the graph $\mathcal{G}$ will be poor and not reflect the underlying structure. As illustrated in Fig. 1(c), a poor-quality graph will lead to the poor smoothness even if the right feature is selected, this insight motivates us to learn graph and features jointly during the learning process.

## 3   Proposed Method

In this paper, we devise a collaborative neural network driven by the Dirichlet Energy for joint feature and graph learning, as illustrated in Fig. 2. Generally, the proposed framework consists of two

---

[2]Note that constant features (if any) will be removed during the feature preprocessing stage.

[3]Note that the similarity matrix $\boldsymbol{S}$ of a $k$-NN graph is not symmetric. When calculating the objective in Eq. (3), we obtain the Laplacian matrix $\boldsymbol{L}_S$ using the symmetrized similarity matrix $\hat{\boldsymbol{S}} = (\boldsymbol{S} + \boldsymbol{S}^\top)/2$.

[4]LS considers both the smoothness and the variance of each feature. However, as we assume that each feature has unit variance according to $\mathbf{1}_n^\top \boldsymbol{x}_i = 0$ and $\boldsymbol{x}_i^\top \boldsymbol{x}_i = 1$, minimizing LS is equivalent to minimizing Eq. (1).

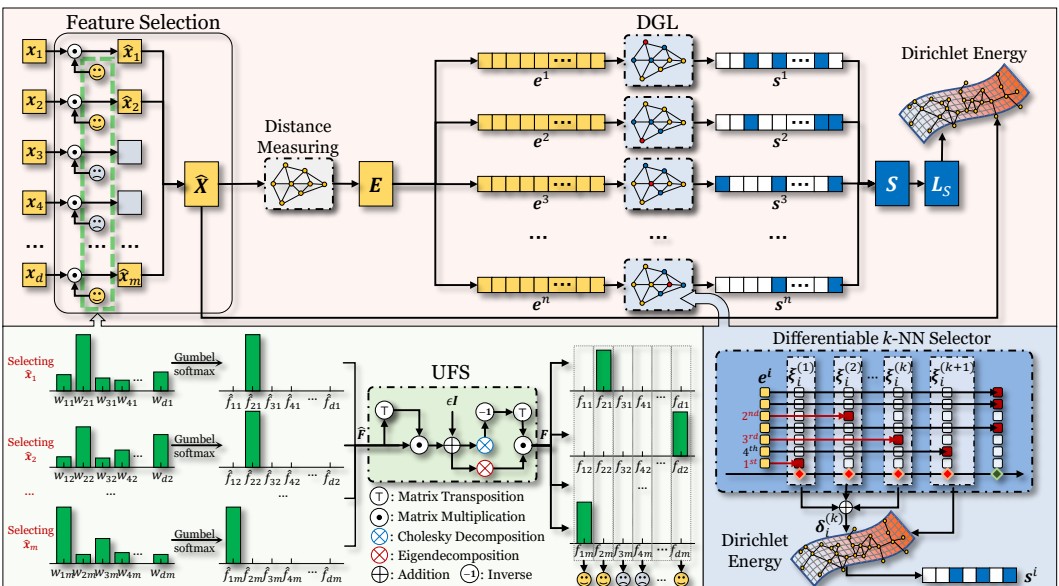

Figure 2: (1) **Top Panel**: Overview of the proposed framework, where smiley faces denote the value 1 representing that the feature is selected, while sad faces denote the value 0 representing that the feature is unused. (2) **Bottom Left Panel**: Illustration of the Unique Feature Selector (UFS), where green bars denote the value distributions of different vectors. (3) **Bottom Right Panel**: Illustration of the Differentiable $k$-NN Graph Learner (DGL), where the "Differentiable $k$-NN Selector" in deep blue shows how to learn $k$ nearest neighbors with the Optimal Transport theory.

modules: the Unique Feature Selector (UFS) and the Differentiable $k$-NN Graph Learner (DGL). At the beginning, the input features $\boldsymbol{X}$ are selected with the learnable feature mask $\boldsymbol{F}$ generated by UFS, which is carefully designed to avoid the duplicate feature selection. Based on the selected data $\hat{\boldsymbol{X}}$, we measure the distances between different samples, and feed the resulting distance vectors of each sample into DGL to learn their $k$ nearest neighbors. The adaptive graph structure and informative features are learned jointly under the Dirichlet Energy, so as to identify the optimal feature subset that effectively captures the underlying data structure.

## 3.1 Unique Feature Selector

Based on the original data $\boldsymbol{X}$, the goal of FS is to identify a feature subset $\hat{\boldsymbol{X}} \in \mathcal{R}^{n \times m}$ from the original features by minimizing a prespecified target $\mathcal{L}_{obj}(\hat{\boldsymbol{X}})$:

$$\min_{\boldsymbol{F}} \mathcal{L}_{obj}(\hat{\boldsymbol{X}}) \quad \text{s.t. } \hat{\boldsymbol{X}} = \boldsymbol{X}\boldsymbol{F}, \boldsymbol{F} \in \{0,1\}^{d \times m}, \boldsymbol{F}^{\top}\boldsymbol{F} = \boldsymbol{I}_m, \tag{2}$$

where $m \leq d$ denotes the number of selected features, and $\boldsymbol{F} \in \mathcal{R}^{d \times m}$ denotes the selection matrix selecting $m$ features from $\boldsymbol{X}$. Different from existing methods that use the reconstruction error as $\mathcal{L}_{obj}(\hat{\boldsymbol{X}})$, in this paper, we utilize the Dirichlet Energy in Eq. (1) for FS as follows:

$$\mathcal{L}_{obj}(\hat{\boldsymbol{X}}) = \sum_{i=1}^{m} \mathcal{L}_{dir}(\hat{\boldsymbol{x}}_i) = \text{tr}(\hat{\boldsymbol{X}}^{\top} \boldsymbol{L}_S \hat{\boldsymbol{X}}). \tag{3}$$

Given the selection number $m$, $\mathcal{L}_{obj}$ updates the network parameters by minimizing the Dirichlet Energy, thereby selecting $m$ features that best reflect the intrinsic structure.

The constraints in problem (2) indicate that an ideal result $\boldsymbol{F}$ should be *exact* and *unique*. *Exact* means the result should exactly be the original features, instead of their linear combinations. *Unique* means each feature should be selected only once under a given number $m$. These two properties require $\boldsymbol{F}$ to be a binary and column-full-rank matrix including $m$ orthogonal one-hot column vectors.

### 3.1.1 Approximating Discrete Feature Selection

It is difficult to learn a discrete $\boldsymbol{F}$ in neural networks due to its non-differentiable property. Inspired by [8], we propose to learn the discrete distribution using the *Gumbel Softmax* [17, 18] technique:

$$\hat{\boldsymbol{f}}_i = \text{softmax}((\log \boldsymbol{w}_i + \boldsymbol{g}_i)/T) \quad \text{with } g_{i,j} = -\log(-\log u_{i,j}), u_{i,j} \sim \text{Uniform}(0,1), \quad (4)$$

where $\boldsymbol{W} = [\boldsymbol{w}_1, \boldsymbol{w}_2, \ldots, \boldsymbol{w}_m]$ denotes a learnable parameter. The random vector $\boldsymbol{g}_i$ consists of $d$ Gumbel-distributed variables $g_{i,j}$, which is generated with $u_{i,j}$ sampled from Uniform distribution. Based on $\boldsymbol{w}_i$ and $\boldsymbol{g}_i$, we obtain the approximated FS vector $\hat{\boldsymbol{f}}_i$ that represents the $i$-th selected feature. The distribution of $\hat{\boldsymbol{f}}_i$ is controlled by a non-negative temperature parameter $T$. A smaller value of parameter $T$ will generate a better approximation of the one-hot vector, but will be more likely to be stuck in a poor local minimum. As suggested in [8], we employ the annealing schedule on $T$ by initializing it with a high value and then gradually decreasing it during the learning process.

### 3.1.2 Selecting Unique Features

Despite having obtained the approximated FS vectors in neural networks, Eq. (4) does not consider the *uniqueness* requirement of FS. This is because Eq. (4) learns each selected feature separately, and does not consider the orthogonal constraint between columns in $\hat{\boldsymbol{F}}$, which is prone to result in the repeated selection of the same features. To address this issue, we develop a unique feature selector (UFS) in Algorithm 1, where $\hat{\boldsymbol{0}} \in \mathcal{R}^{(d-m)\times m}$ denotes the zero matrix. First, we add a small enough perturbation $\epsilon \boldsymbol{I}_m (\epsilon > 0)$ on $\hat{\boldsymbol{F}}^\top \hat{\boldsymbol{F}}$. Next, we perform

---

**Algorithm 1** UFS

1: **procedure** UFS($\hat{\boldsymbol{F}}, \epsilon$)

2: $\quad\quad \boldsymbol{P}\boldsymbol{\Lambda}\boldsymbol{P}^\top = \hat{\boldsymbol{F}}^\top \hat{\boldsymbol{F}} + \epsilon \boldsymbol{I}_m$

3: $\quad\quad \boldsymbol{L}\boldsymbol{L}^\top = \hat{\boldsymbol{F}}^\top \hat{\boldsymbol{F}} + \epsilon \boldsymbol{I}_m$

4: $\quad\quad \boldsymbol{F} = \begin{bmatrix} \boldsymbol{\Lambda}^{1/2}\boldsymbol{P}^\top \\ \hat{\boldsymbol{0}} \end{bmatrix} (\boldsymbol{L}^{-1})^\top$

5: $\quad\quad$ **return** $\boldsymbol{F}$

6: **end procedure**

---

the eigendecomposition (line 2) and the Cholesky decomposition (line 3) on the perturbed result respectively, and correspondingly obtain the diagonal matrix $\boldsymbol{\Lambda} \in \mathcal{R}^{m\times m}$, the orthogonal matrix $\boldsymbol{P} \in \mathcal{R}^{m\times m}$, and the lower triangle matrix $\boldsymbol{L} \in \mathcal{R}^{m\times m}$. Based on $\boldsymbol{\Lambda}$, $\boldsymbol{P}$, and $\boldsymbol{L}$, we obtain the selection matrix $\boldsymbol{F}$ in line 4 and have the following conclusion:

**Proposition 3.1.** *Given any real matrix $\hat{\boldsymbol{F}} \in \mathcal{R}^{d\times m}$, one can always generate a column-orthogonal matrix $\boldsymbol{F}$ through Algorithm 1.*

The proof of Proposition 3.1 is provided in Appendix S1. On the one hand, the small perturbation $\epsilon \boldsymbol{I}_m$ guarantees the column-full-rank property of $\boldsymbol{F}$, thereby avoiding the duplicate selection results. On the other hand, the orthogonality property in Proposition 3.1 facilitates the approximation of discrete FS based on the matrix $\hat{\boldsymbol{F}}$.[5] We verify the efficacy of UFS in Section 4.2.

## 3.2 Differentiable $k$-NN Graph Learner

The existence of noise and irrelevant features may negatively affect the quality of the constructed graph. As depicted in Fig. 1, a low-quality graph structure can significantly perturb the smoothness of features and undermine the performance of feature selection. Hence, we propose to learn an adaptive graph during the learning process using the selected features.

### 3.2.1 Learning an Adaptive $k$-NN Graph Using Dirichlet Energy

Considering the objective function in Eq. (3), a natural way is to learn the similarity matrix $\boldsymbol{S}$ based on the Dirichlet Energy in $\mathcal{L}_{obj}$. However, this may yield a trivial solution where, for sample $\boldsymbol{x}^i$, only the nearest data point can serve as its neighbour with probability 1, while all the other data points will not be its neighbours. To avoid this trivial solution, we propose to learn an adaptive graph by incorporating the *Tikhonov regularization* [19] of $\boldsymbol{S}$ into the Dirichlet Energy:

$$\min_{\boldsymbol{S}} \text{tr}(\hat{\boldsymbol{X}}^\top \boldsymbol{L}_S \hat{\boldsymbol{X}}) + \frac{\alpha}{2}\|\boldsymbol{S}\|_F^2 \quad \text{s.t. } \boldsymbol{S}\boldsymbol{1}_n = \boldsymbol{1}_n, s_{i,j} \geq 0, s_{i,i} = 0, \quad (5)$$

where $\alpha$ denotes the trade-off parameter between the Dirichlet Energy and the Tikhonov regularization. Note that each row $\boldsymbol{s}^i$ in $\boldsymbol{S}$ can be solved separately, instead of tuning $\alpha$ manually, we model $\alpha$ as

---

[5]In practice, we calculate $\boldsymbol{F}$ with $\boldsymbol{F} = \hat{\boldsymbol{F}}(\boldsymbol{L}^{-1})^\top$ without eigendecomposition to achieve discrete FS vectors. We provide a discussion in Appendix S2 to explain this treatment.

a sample-specific parameter $\alpha_i$ and determine it algorithmically, which plays an important role in learning $k$ nearest neighbors for each sample. Based on problem (5), we define the distance matrix $\boldsymbol{E}$ with its entries being $e_{i,j} = \|(\hat{\boldsymbol{x}}^i - \hat{\boldsymbol{x}}^j)\|_2^2$, then we solve each row $\boldsymbol{s}^i$ in problem (5) separately as

$$\min_{\boldsymbol{s}^i} \frac{1}{2}\|\boldsymbol{s}^i + \frac{\boldsymbol{e}^i}{2\alpha_i}\|_2^2 \quad \text{s.t. } \boldsymbol{s}^i \mathbf{1}_n = 1, s_{i,j} \geq 0, s_{i,i} = 0. \tag{6}$$

Problem (6) can be solved easily by constructing the Lagrangian function and then using the Karush-Kuhn-Tucker(KKT) conditions [20]. By doing so, we obtain the solution of $s_{i,j}$ as

$$s_{i,j} = \left(\frac{1}{k} + \frac{1}{k}\frac{\boldsymbol{e}^i \boldsymbol{\delta}_i^{(k)}}{2\alpha_i} - \frac{e_{i,j}}{2\alpha_i}\right)_+ \quad \text{with} \quad \delta_{i,j}^{(k)} = \text{Bool}(e_{i,j} \leq e_{i,\sigma_k}), \tag{7}$$

where $\boldsymbol{\sigma} = [\sigma_1, \ldots, \sigma_n]$ denotes the sorting permutation over $\boldsymbol{e}^i$, i.e. $e_{i,\sigma_1} \leq \cdots \leq e_{i,\sigma_n}$ and $\boldsymbol{\delta}_i^{(k)}$ denotes the selection vector identifying the $k$ minimal values in $\boldsymbol{e}^i$.

Recall that we aim to learn $k$ nearest neighbors for each sample, which implies that there are only $k$ nonzero elements in $\boldsymbol{s}^i$ corresponding to the nearest neighbors. To this end, we determine the trade-off parameters $\alpha_i$ such that $s_{i,\sigma_k} > 0$ and $s_{i,\sigma_{k+1}} \leq 0$. Then we have:

$$\frac{1}{2}(k\boldsymbol{e}^i \boldsymbol{\xi}_i^{(k)} - \boldsymbol{e}^i \boldsymbol{\delta}_i^{(k)}) < \alpha_i \leq \frac{1}{2}(k\boldsymbol{e}^i \boldsymbol{\xi}_i^{(k+1)} - \boldsymbol{e}^i \boldsymbol{\delta}_i^{(k)}) \quad \text{with} \quad \xi_{i,j}^{(k)} = \text{Bool}(e_{i,j} = e_{i,\sigma_k}), \tag{8}$$

where $\boldsymbol{\xi}_i^{(k)}$ denotes an indicator vector identifying the $k$-th minimal value in $\boldsymbol{e}^i$. Setting $\alpha_i$ as the maximum and substituting it into Eq. (7), we obtain the final solution as:

$$s_{i,\sigma_j} = \frac{\boldsymbol{e}^i \boldsymbol{\xi}_i^{(k+1)} - e_{i,\sigma_j}}{k\boldsymbol{e}^i \boldsymbol{\xi}_i^{(k+1)} - \boldsymbol{e}^i \boldsymbol{\delta}_i^{(k)}} \cdot \text{Bool}(1 \leq j \leq k). \tag{9}$$

The detailed derivation of solution (9) can be found in Appendix S3. We note that the formulation in problem (6) bears similarity to CLR proposed in [21]. In Appendix S4, we discuss the connection between our method and CLR, and highlight the differences between the two w.r.t. the feature utilization and the sorting operation. Remarkably, the $k$-NN can be obtained easily in CLR using off-the-shelf sorting algorithms, which is not the case for neural networks due to the non-differentiability of sorting algorithms. To address this issue, we propose to transform the $k$-NN selection into a differentiable operator utilizing the *Optimal Transport* (OT) [22] technique as follows.

### 3.2.2 Differentiable $k$-NN Selector

Let $\boldsymbol{\mu} = [\mu_1, \mu_2, \cdots, \mu_{n_1}]^\top$ and $\boldsymbol{\nu} = [\nu_1, \nu_2, \cdots, \nu_{n_2}]^\top$ be two discrete probability distributions defined on the supports $\mathcal{A} = \{\mathfrak{a}_i\}_{i=1}^{n_1}$ and $\mathcal{B} = \{\mathfrak{b}_j\}_{j=1}^{n_2}$ respectively . The goal of OT is to find an optimal transport plan $\boldsymbol{\Gamma} \in \mathcal{R}^{n_1 \times n_2}$ between $\mathcal{A}$ and $\mathcal{B}$ by minimizing the following transport cost:

$$\min_{\boldsymbol{\Gamma}}\langle \boldsymbol{C}, \boldsymbol{\Gamma}\rangle, \quad \text{s.t. } \boldsymbol{\Gamma}\mathbf{1}_{n_2} = \boldsymbol{\mu}, \boldsymbol{\Gamma}^\top \mathbf{1}_{n_1} = \boldsymbol{\nu}, \Gamma_{i,j} \geq 0, \tag{10}$$

where $\boldsymbol{C} \in \mathcal{R}^{n_1 \times n_2}$ denotes the cost matrix with $c_{i,j} = h(\mathfrak{a}_i - \mathfrak{b}_j) > 0$ being the transport cost from $\mathfrak{a}_i$ to $\mathfrak{b}_j$. It is widely known that the solution of the OT problem between two discrete univariate measures boils down to the sorting permutation [23–25]. As stated in [25], if $h$ is convex, the optimal assignment can be achieved by assigning the smallest element in $\mathcal{A}$ to $\mathfrak{b}_1$, the second smallest to $\mathfrak{b}_2$, and so forth, which eventually yields the sorting permutation of $\mathcal{A}$.

Given a distance vector $\boldsymbol{e}$, to learn selection vectors $\boldsymbol{\delta}^{(k)}$ and $\boldsymbol{\xi}^{(k+1)}$, we set $\mathcal{A} = \boldsymbol{e}$, and $\mathcal{B} = [0, 1, \ldots, k+1]$, and define $\boldsymbol{\mu}, \boldsymbol{\nu}$, and $c_{ij}$ as

$$\mu_i = \frac{1}{n}, \quad \nu_j = \begin{cases} 1/n, & 1 \leq j \leq k+1 \\ (n-k-1)/n, & j = k+2 \end{cases}, \quad c_{ij} = (\mathfrak{a}_i - \mathfrak{b}_j)^2 = (e_i - j + 1)^2. \tag{11}$$

The optimal transport plan of problem (10) assigns the $i$-th smallest value $e_{\sigma_i}$ to $\mathfrak{b}_i$ if $1 \leq i \leq k+1$, and assigns the remaining $n - k - 1$ values in $\boldsymbol{e}$ to $\mathfrak{b}_{k+2}$. Namely,

$$\Gamma_{\sigma_i,j} = \begin{cases} 1/n, & if \ (1 \leq i \leq k+1 \ and \ j = i) \ or \ (k+1 < i \leq n \ and \ j = k+2) \\ 0, & if \ (1 \leq i \leq k+1 \ and \ j \neq i) \ or \ (k+1 < i \leq n \ and \ j \neq k+2) \end{cases}. \tag{12}$$

Table 1: Details of real-world data.

| Type | Dataset | #Samples | #Features | #Classes | Type | Dataset | #Samples | #Features | #Classes |
|------|---------|----------|-----------|----------|------|---------|----------|-----------|----------|
| Text | PCMAC [28] | 1943 | 3289 | 2 | Artificial | Madelon [29] | 2600 | 500 | 2 |
| | GLIOMA [30] | 50 | 4434 | 4 | | COIL-20 [31] | 1440 | 1024 | 20 |
| | LUNG [32] | 203 | 3312 | 5 | | Yale [33] | 165 | 1024 | 15 |
| Biological | PROSTATE [34] | 102 | 5966 | 2 | Image | Jaffe [35] | 213 | 676 | 10 |
| | SRBCT [36] | 83 | 2308 | 4 | | PIX10 [14] | 100 | 10000 | 10 |
| | SMK [37] | 187 | 19993 | 2 | | warpPIE10P [38] | 210 | 2420 | 10 |

Given a sample $p$, once we obtain the optimal transport assignment $\boldsymbol{\Gamma}$ based on $\boldsymbol{e}^p$, we calculate the variables $\boldsymbol{\delta}_p^{(k)}$ and $\boldsymbol{\xi}_p^{(k+1)}$ as follows:

$$\boldsymbol{\delta}_p^{(k)} = n \sum_{i=1}^{k} \boldsymbol{\Gamma}_i, \quad \boldsymbol{\xi}_p^{(k+1)} = n\boldsymbol{\Gamma}_{k+1}, \tag{13}$$

where $\boldsymbol{\Gamma}_i$ and $\boldsymbol{\Gamma}_{k+1}$ denote the $i$-th and the $(k+1)$-th column of $\boldsymbol{\Gamma}$, respectively. However, problem (10) is still non-differentiable. To address this issue, we consider the following entropy regularized OT problem:

$$\min_{\boldsymbol{\Gamma}} \langle \boldsymbol{C}, \boldsymbol{\Gamma} \rangle + \gamma \sum_{i,j} \Gamma_{i,j} \log \Gamma_{i,j} \quad \text{s.t. } \boldsymbol{\Gamma} \mathbf{1}_{k+2} = \boldsymbol{\mu}, \boldsymbol{\Gamma}^\top \mathbf{1}_n = \boldsymbol{\nu}, \Gamma_{i,j} \geq 0, \tag{14}$$

where $\gamma$ is a hyperparameter. The differentiability of problem (14) has been proven using the implicit function theorem (see [26, Theorem 1]). Note that a smaller $\gamma$ yields a better approximation to the original solution in problem (10), but may compromise the differentiability of problem (14) [25]. Problem (14) can be solved efficiently using the iterative Bregman projections algorithm [27], the details of which are provided in Appendix S5.

# 4 Experiments

Our experiments fall into three parts: (1) *Toy Experiments*: First, we verify the FS ability and the graph learning ability of the proposed method on synthetic datasets. (2) *Quantitative Analysis*: Next, we compare the performance of selected features in various downstream tasks on real-world datasets and compare our method with other unsupervised FS methods. (3) *Ablation Study*: Finally, we verify the effect of UFS and DGL by testing the performance of the corresponding ablated variants. We also provide the sensitivity analysis in Appendix S6.6. The implementation details of all experiments can be found in Appendix S6.1.

## 4.1 Datasets

For the toy experiments, we generate three 20-dimensional datasets named Blobs, Moons, and Circles (see Appendix S6.1.1 for generation details). On top of that, we evaluate the proposed method on twelve real-world datasets that include text, biological, image, and artificial data. Table 1 exhibits the details of these datasets, which include many high-dimensional datasets to test the performance of our method. We standardize all features to zero means and normalize them with the standard deviation.

## 4.2 Toy Experiments

In this section, we consider three synthetic binary datasets with increasing difficulty in separating different classes. The first two dimensions in each dataset contain useful features that indicate the underlying structure, while the remaining 18 dimensions are random noise sampled from $\mathcal{N}(0, 1)$. The presence of noise obscures the inherent structure of the data, which makes the graph learning process highly challenging. To see this, we generate 3-D plots of each dataset using the useful features and one noise feature, along with their 2-D projections on each plane, which are shown in Fig. 3(a). We can see that the noise blurs the boundary of different classes, especially in Moons and Circles. In addition, we used a heat kernel (abbreviated as Heat) with $\sigma = 1$ to learn the 5-NN graph on 20-dimensional features, as shown in Fig. 3(b). We can see that the heavy noise obscures the underlying structure of data points, resulting in a chaotic graph outcome.

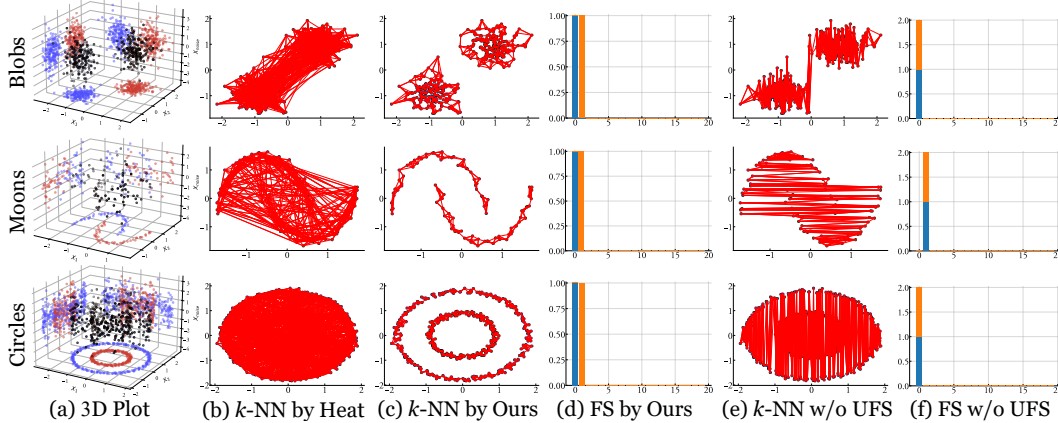

|  | (a) 3D Plot | (b) $k$-NN by Heat | (c) $k$-NN by Ours | (d) FS by Ours | (e) $k$-NN w/o UFS | (f) FS w/o UFS |

Figure 3: Toy results on synthetic datasets, where higher similarities are presented with thicker connections in $k$-NN graphs, and we only present the connections to 5-NN for each sample. Blue bars and orange bars represent the distribution of $\boldsymbol{f}_1$ and $\boldsymbol{f}_2$ in the FS matrix $\boldsymbol{F}$, respectively.

**Results.** We test our method on toy datasets for selecting $m = 2$ target features. The results are presented in Fig. 3(c) and Fig. 3(d), which demonstrate the success of our method in learning target features and intrinsic structures simultaneously. Moreover, it can be seen from Fig. 3(d) that the proposed network obtains the approximately discrete FS vectors.

**Learning Without Unique Feature Selector.** In addition, we conduct an ablation study by removing the UFS module from the network and updating $\boldsymbol{F}$ using Eq. (4) only. The results are shown in Fig. 3(e) and Fig. 3(f), where we can see that, without UFS, the ablated model repeatedly selects the same feature on all datasets. It is also noteworthy that the nodes in the graph are mostly connected either horizontally or vertically, indicating the effectiveness of DGL in learning the local structure solely based on the single selected feature.

## 4.3 Quantitive Analysis

**Experimental Settings.** In this section, we evaluate our method on real-world data. We partition each dataset into training data and testing data using an 8:2 ratio and identify useful features using training data. We then evaluate the performance of selected features on three downstream tasks: (1) *Classification Accuracy*: We train a random forest (RF) [39] classifier with 1000 trees using selected features and evaluate the prediction accuracy on the testing data. (2) *Clustering Accuracy*: We cluster the testing set with selected features using $k$-means [40], where the cluster number is set to #Classes. Then we align the results with true labels and calculate the accuracy. (3) *Reconstruction RMSE*: We build a 1-hidden-layer network with ReLU activation to reconstruct the original data using selected features. The hidden dimension is set to $3m/2$ except for AllFea, where the hidden size is set to $d$. The network is learned on the training set with selected features, and evaluated on the testing set using root mean square error (RMSE) normalized by $d$.

**Competing methods.** We compare our methods with four deep methods (**CAE** [8], **DUFS** [14], **WAST** [41], **AEFS** [9]) and three classical methods (**LS** [13], **RSR** [42], **UDFS** [43]). Besides, we use all features (**AllFea**) as the baseline. To evaluate the performance of each FS method on downstream tasks, we average the results over 10 random runs with the feature number $m$ varied in $\{25, 50, 75, 100, 150, 200, 300\}$ except for Madelon, where $m$ is varied in $\{5, 10, 15, 20\}$ since Madelon consists of only 20 useful features [29]. Appendix S6.1 provides details of the overall evaluation workflow, including the implementation and the parameter selection of each method.

**Results.** Similar to [14], we present the best result w.r.t. $m$ in Table 2, the standard deviations is provided in Appendix S6.2. We also present some reconstruction results on PIX10 by our method in Appendix S6.3. From Table 2, we find that: (1) Our method generally achieves the best performance in all three tasks, indicating that our method selects more useful features. (2) In particular, we beat DUFS in all Classification and Clustering tasks, as well as most cases of the Reconstruction tasks. Recall that DUFS also selects features based on the Dirichlet Energy, this result shows that the model (5) in our method explores a superior graph structure compared to the traditional Heat method. (3)

Table 2: Results in downstream tasks over 10 runs on optimal $m$ that is shown in the bracket. "Cla.", "Clu." and "Rec." are short for classification, clustering, and reconstruction, respectively.

| Task | Dataset | CAE | DUFS | WAST | AEFS | LS | RSR | UDFS | Our | AllFea |
|---|---|---|---|---|---|---|---|---|---|---|
| Cla. (ACC↑) | Madelon | 0.65 (15) | 0.52 (20) | 0.87 (15) | 0.87 (10) | 0.51 (20) | 0.83 (10) | 0.74 (20) | **0.90 (20)** | 0.73 |
| | PCMAC | 0.79 (50) | 0.77 (300) | 0.83 (300) | 0.71 (300) | 0.68 (300) | 0.91 (300) | 0.87 (300) | 0.79 (300) | **0.93** |
| | COIL-20 | **1.00 (300)** | **1.00 (300)** | **1.00 (200)** | **1.00 (300)** | 0.98 (300) | **1.00 (200)** | **1.00 (300)** | **1.00 (200)** | 1.00 |
| | Yale | 0.72 (300) | 0.72 (300) | 0.70 (200) | 0.70 (100) | 0.67 (300) | 0.72 (200) | 0.71 (200) | **0.81 (300)** | 0.75 |
| | Jaffe | 0.97 (50) | 0.97 (50) | 0.97 (25) | 0.97 (100) | 0.97 (300) | 0.98 (300) | 0.98 (75) | **1.00 (150)** | 0.98 |
| | PIX10 | 0.99 (25) | 0.98 (50) | 0.99 (150) | 0.98 (100) | 0.97 (100) | 0.97 (50) | 0.97 (100) | **1.00 (25)** | 0.97 |
| | warpPIE10P | 0.96 (100) | 0.96 (300) | 0.95 (300) | 0.98 (75) | 0.96 (300) | 0.98 (300) | 0.97 (200) | **0.99 (300)** | 0.98 |
| | GLIOMA | 0.68 (200) | 0.66 (300) | 0.72 (100) | 0.63 (300) | 0.67 (200) | 0.67 (300) | 0.68 (75) | **0.81 (300)** | 0.72 |
| | LUNG | 0.86 (100) | 0.87 (150) | 0.87 (300) | 0.87 (300) | 0.92 (300) | 0.93 (300) | 0.91 (300) | **0.94 (50)** | 0.90 |
| | PROSTATE | 0.88 (150) | 0.81 (150) | 0.81 (300) | 0.81 (150) | 0.88 (300) | **0.90 (150)** | 0.89 (300) | 0.89 (300) | 0.89 |
| | SRBCT | 0.99 (300) | 0.94 (150) | 0.98 (300) | 0.95 (200) | 0.98 (300) | **1.00 (200)** | **1.00 (150)** | 0.98 (150) | 0.99 |
| | SMK | 0.67 (200) | 0.66 (50) | 0.65 (50) | 0.66 (150) | 0.72 (300) | 0.72 (75) | 0.73 (100) | **0.75 (200)** | 0.68 |
| | Average ranking | 4.7 | 5.9 | 4.9 | 5.5 | 6.3 | 2.8 | 3.3 | 1.8 | 3.2 |
| | # Top-1 | 1 | 1 | 1 | 1 | 0 | 3 | 3 | 9 | 2 |
| Clu. (ACC↑) | Madelon | 0.60 (15) | 0.52 (15) | 0.52 (15) | 0.55 (5) | 0.52 (20) | **0.61 (15)** | 0.57 (20) | 0.60 (20) | 0.58 |
| | PCMAC | 0.52 (150) | 0.51 (25) | 0.51 (25) | 0.51 (100) | 0.52 (75) | 0.51 (25) | 0.51 (150) | **0.53 (200)** | 0.51 |
| | COIL-20 | **0.69 (100)** | 0.59 (150) | 0.65 (200) | 0.60 (300) | 0.53 (300) | 0.59 (300) | 0.6 (300) | 0.66 (200) | 0.63 |
| | Yale | 0.55 (100) | 0.55 (150) | 0.56 (100) | 0.54 (300) | 0.58 (300) | 0.60 (300) | 0.58 (200) | **0.62 (300)** | 0.61 |
| | Jaffe | 0.85 (200) | 0.80 (300) | 0.83 (300) | 0.80 (300) | 0.76 (300) | 0.83 (150) | 0.83 (200) | **0.87 (200)** | 0.82 |
| | PIX10 | 0.86 (200) | 0.79 (150) | 0.85 (150) | 0.79 (300) | 0.85 (75) | 0.72 (300) | 0.81 (200) | **0.87 (300)** | 0.78 |
| | warpPIE10P | 0.55 (75) | 0.42 (300) | 0.44 (50) | 0.53 (25) | 0.55 (200) | **0.57 (75)** | 0.49 (25) | 0.51 (25) | 0.45 |
| | GLIOMA | 0.69 (50) | 0.65 (50) | 0.65 (25) | 0.62 (300) | 0.63 (300) | 0.65 (150) | 0.68 (75) | **0.75 (75)** | 0.62 |
| | LUNG | 0.64 (150) | 0.64 (100) | 0.62 (75) | 0.66 (150) | 0.65 (200) | 0.69 (300) | 0.61 (300) | **0.72 (150)** | 0.69 |
| | PROSTATE | 0.64 (25) | 0.59 (150) | 0.58 (300) | 0.59 (25) | **0.71 (25)** | 0.63 (25) | 0.69 (25) | 0.68 (50) | 0.64 |
| | SRBCT | **0.76 (150)** | 0.54 (75) | 0.56 (200) | 0.57 (200) | 0.56 (200) | 0.60 (150) | 0.59 (150) | 0.63 (50) | 0.52 |
| | SMK | 0.60 (300) | 0.58 (25) | 0.58 (25) | 0.58 (50) | 0.59 (50) | 0.59 (50) | 0.61 (300) | **0.64 (25)** | 0.60 |
| | Average ranking | 2.8 | 6.6 | 5.7 | 6 | 5 | 4 | 4.3 | 1.8 | 5.1 |
| | # Top-1 | 2 | 0 | 0 | 0 | 1 | 2 | 0 | 7 | 0 |
| Rec. (RMSE↓) | Madelon | 0.99 (20) | 0.99 (20) | 0.98 (20) | 0.98 (20) | 0.99 (20) | 0.98 (20) | 0.98 (20) | 0.98 (10) | **0.28** |
| | PCMAC | 1.14 (25) | 1.03 (25) | 1.05 (25) | 1.05 (25) | 1.04 (25) | 1.17 (50) | 1.07 (25) | 1.30 (25) | **0.78** |
| | COIL-20 | 0.48 (300) | 0.41 (300) | 0.43 (300) | 0.41 (300) | 0.48 (300) | 0.45 (300) | 0.41 (300) | 0.38 (300) | **0.27** |
| | Yale | 0.63 (300) | 0.56 (300) | 0.56 (300) | 0.56 (300) | 0.78 (300) | 0.57 (300) | 0.60 (200) | 0.54 (300) | **0.50** |
| | Jaffe | 0.31 (300) | 0.25 (300) | 0.26 (300) | 0.25 (300) | 0.33 (300) | 0.26 (300) | 0.25 (300) | **0.22 (300)** | 0.23 |
| | PIX10 | 0.49 (300) | 0.43 (300) | 0.46 (300) | 0.43 (300) | 0.63 (50) | 0.46 (300) | 0.47 (300) | **0.39 (300)** | 1.20 |
| | warpPIE10P | 0.30 (300) | 0.26 (300) | 0.27 (300) | 0.26 (300) | 0.45 (300) | 0.27 (300) | 0.26 (300) | **0.25 (300)** | 0.25 |
| | GLIOMA | 0.74 (300) | 0.71 (300) | 0.71 (300) | 0.72 (300) | 0.71 (300) | 0.72 (300) | 0.72 (300) | **0.69 (300)** | 1.86 |
| | LUNG | 0.94 (300) | **0.77 (300)** | **0.77 (300)** | 0.78 (300) | 0.82 (300) | 0.81 (300) | 0.79 (300) | 0.80 (300) | 0.82 |
| | PROSTATE | 1.00 (300) | 0.78 (300) | 0.77 (300) | 0.77 (300) | 0.72 (300) | 0.74 (300) | **0.71 (300)** | 0.73 (300) | 1.48 |
| | SRBCT | 0.84 (300) | 0.77 (300) | 0.77 (300) | 0.78 (300) | 0.80 (300) | 0.79 (300) | 0.78 (300) | **0.75 (300)** | 0.83 |
| | SMK | 0.98 (25) | **0.68 (300)** | **0.68 (300)** | **0.68 (300)** | 0.78 (300) | 0.78 (300) | 0.73 (300) | 0.69 (300) | 4.32 |
| | Average ranking | 7.9 | 3 | 3.5 | 3.2 | 6.4 | 5.5 | 4.1 | 2.7 | 4.8 |
| | # Top-1 | 0 | 2 | 2 | 1 | 0 | 0 | 1 | 5 | 5 |

Table 3: Results in ablation studies, where "w/o" is short for "without".

| Task | Method | Madelon | PCMAC | Jaffe | PIX10 | GLIOMA | PROSTATE |
|---|---|---|---|---|---|---|---|
| Effect of FS (Clu. with SC) | Heat | 0.54±0.01 | **0.51±0.00** | 0.60±0.13 | 0.41±0.06 | 0.48±0.05 | 0.55±0.02 |
| | DGL only | 0.50±0.00 | 0.50±0.00 | 0.57±0.05 | 0.76±0.09 | 0.55±0.04 | 0.56±0.07 |
| | Our | **0.58±0.01** | **0.51±0.00** | **0.80±0.07** | **0.77±0.04** | **0.59±0.04** | **0.65±0.02** |
| Effect of DGL (Cla. with RF) | w/o DGL | 0.51±0.02 | 0.72±0.02 | 0.97±0.01 | 0.98±0.03 | 0.66±0.10 | 0.81±0.04 |
| | Our | **0.90±0.01** | **0.79±0.01** | **1.00±0.01** | **1.00±0.00** | **0.81±0.09** | **0.89±0.05** |

In classification and clustering, the best performance is mostly achieved by FS methods with fewer features, which verifies the necessity of FS. (4) AllFea achieves five optimums in Reconstruction, which is not surprising since, theoretically, AllFea can be projected to original features with an identity matrix. However, in biological data, the best reconstruction results are achieved by FS methods, probably because the high-dimensional data leads to overfitting in networks. (5) It is noteworthy that the reconstruction of Madelon poses a significant challenge for FS methods, indicating the difficulty of reconstructing noise even using useful features. This observation supports our claim regarding the lack of reasonability in selecting features based on the reconstruction performance in Section 1.

## 4.4 Ablation Study

In this experiment, we demonstrate the efficacy of the UFS and the DGL modules through ablation studies on six datasets: Madelon, PCMAC, Jaffe, PIX10, GLIOMA, and PROSTATE.

**Effect of FS.** Recall that we have demonstrated the efficacy of UFS in Fig. 3. To further verify the efficacy of FS in graph learning, we remove the entire FS module from the framework and learn the graph using all features based on DGL. We also compare the graph learning result using Heat. We

cluster the obtained graphs with the spectral clustering (SC) method to verify their qualities. We tune the parameter $\sigma$ of Heat in $\{1, 2, \ldots, 5\}$, and fix $k = 5$ for our method and the variant. The results are shown in Table 3, which shows that FS has a positive effect on graph learning compared with "DGL only". Besides, in Appendix S6.4, we visualize the learned graph on COIL-20 and Jaffe using t-SNE, which shows that using fewer features, we achieve separable graphs that contain fewer inter-class connections than other methods.

**Effect of DGL.** To verify the efficacy of DGL, we remove it from the model and learn the ablated variant with a fixed graph learned by Heat. Similar to Section 4.3, we first learn selected features using competing method, then evaluate the features in downstream tasks. We present the classification result in Table 3, and leave the other results in Appendix S6.5 due to limited space. We can see that our method significant outperforms the ablated variant, especially in Madelon. This is probably because the noise undermine the graph structure and disrupt the learning of informative features.

## 5  Discussion

**Conclusion.** This paper proposes a deep unsupervised FS method that learns informative features and $k$-NN graph jointly using the Dirichlet Energy. The network is fully differentiable and all modules are developed algorithmically to present versatility and interpretability. We demonstrate the performance of our method with extensive experiments on both synthetic and real-world datasets.

**Broader Impact.** This paper presents not only an effective deep FS method, but also a differentiable $k$-NN graph learning strategy in the context of deep learning. This technique is particularly useful for end-to-end learning scenarios that require graph learning during the training process. And we do notice this practical need in existing literature, see [44] for example. We believe our study will inspire researchers who work on the dimensionality reduction and graph-related researches.

**Limitations.** The major limitation of the proposed method is the lack of scalability, for which we do not evaluate our method on large datasets. This is because problem (14) requires an iterative solution, requiring storage of all intermediate results for back-propagation. While literature [26] proposes a memory-saving approach by deriving the expression of the derivative of $\boldsymbol{\Gamma}$ mathematically (see [26, Section 3]), it still requires at least $\mathcal{O}(nk)$ space to update all intermediate variables to learn $k$ nearest neighbors for a singe sample, which results in a $\mathcal{O}(n^2 k)$ space complexity to learn for all $n$ samples. This is a huge memory cost on large datasets. Although learning in batch seems to be the most straightforward solution, in our method, the neighbours of each sample are determined based on the global information of $\boldsymbol{L}_S$, which has an $n \times n$ size. This requires to load the entire batch's information during each iteration, for which we cannot employ subgraph sampling as other graph learning methods did to mitigate memory overhead. Another limitation of the proposed method is the low computational speed, as it is reported that the OT-based sorting can be slow [45].

The future developments of the proposed method are twofold. First, we will try more differentiable sorting algorithms to enhance computational speed. For example, reference [45] proposes to construct differentiable sorting operators as projections onto the permutahedron, which achieves a $\mathcal{O}(n \log n)$ forward complexity and a $\mathcal{O}(n)$ backward complexity. Second, due to the large cost of the global relationship in $\boldsymbol{L}_S$, we are considering adopting a bipartite graph [16, 46] to make batch learning feasible. This graph introduces a small number of anchor points, which are representative of the entire feature space. By doing this, smoothness can be measured based on the distance between samples to anchors, for which sample-to-sample relationships are no longer needed and the batch learning is enabled. It is worth noting that this idea is still in its conceptual stage, and we will explore its feasibility in upcoming research.

## Acknowledgments

This work was supported in part by the National Natural Science Foundation of China under Grant 62276212 and Grant 61872190, in part by the National Key Research and Development Program of China under Grant 2022YFB3303800, and in part by the Key Research and Development Program of Jiangsu Province under Grant BE2021093.

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

# S1 Proof of Proposition 3.1

**Proposition 3.1.** *Given any real matrix $\hat{F} \in \mathcal{R}^{d \times m}$, one can always generate a column-orthogonal matrix $F$ through Algorithm 1.*

*Proof.* We begin our proof by showing the feasibility of Algorithm 1 for any real matrix $\hat{F}$, as the eigendecomposition, Cholesky decomposition, and inverse mentioned in the algorithm are subject to specific conditions. For simplicity, we represent $A = \hat{F}^\top \hat{F} + \epsilon I_m$, with $\epsilon > 0$. Note that for any nonzero real column vector $z \in \mathcal{R}^m$, we have

$$z^\top A z = z^\top (\hat{F}^\top \hat{F} + \epsilon I_m) z = (\hat{F}z)^\top (\hat{F}z) + \epsilon z^\top z = \sum_{i=1}^{d}(\hat{f}^i z)^2 + \epsilon \sum_{i=1}^{m} z_i^2 > 0. \qquad (S1)$$

Hence, the matrix $A$ is positive-definite and can be eigendecomposed as $A = P \Lambda P^{-1}$, where $P \in \mathcal{R}^{m \times m}$ is the square matrix whose $i$-th column $p_i$ is the eigenvector of $A$ and $\Lambda \in \mathcal{R}^{m \times m}$ is the diagonal matrix whose diagonal entries are the corresponding eigenvalues. Moreover, it is easy to show that $A$ is symmetric, for which we have $P^\top = P^{-1}$. Therefore, we prove that $A$ can be decomposed as $A = P \Lambda P^\top$ (*line 2* in Algorithm 1).

Since $A$ is symmetric and positive-definite, we will be able to perform Cholesky decomposition on $A$ as $LL^\top = A$ (*line 3* in Algorithm 1), which yields a lower triangular matrix $L \in \mathcal{R}^{m \times m}$ whose diagonal entries are all real and positive. This means that the determinant of $L$ is larger than zero and $L$ is invertible, which provides the feasibility of $L^{-1}$ (*line 4* in Algorithm 1). Consequently, the feasibility of Algorithm 1 for any real matrix $\hat{F}$ is proved.

Next, we show that $F$ is a column-orthogonal matrix. We denote $Q = \begin{bmatrix} \Lambda^{1/2} P^\top \\ \hat{0} \end{bmatrix}$ and have:

$$Q^\top Q = \begin{bmatrix} P \Lambda^{1/2}, \ \hat{0}^\top \end{bmatrix} \begin{bmatrix} \Lambda^{1/2} P^\top \\ \hat{0} \end{bmatrix} = P \Lambda P^\top = A = LL^\top. \qquad (S2)$$

Then we prove the orthogonality of the matrix $F$ as follows:

$$\begin{aligned} F^\top F &= L^{-1} Q^\top Q (L^{-1})^\top \\ &= L^{-1} LL^\top (L^{-1})^\top \\ &= L^{-1} LL^\top (L^\top)^{-1} \\ &= I_m \end{aligned} \qquad (S3)$$

The proof is completed. $\qquad \square$

# S2 Discussion on Algorithm 1

Although Algorithm 1 theoretically guarantees the orthogonality of the selection matrix $F$, utilizing this algorithm directly would bring us back to the problem of how to choose features and how to obtain discrete results. On one hand, the non-uniqueness of eigendecomposition in line 2 prevents us from ensuring the discrete properties of matrix $F$. On the other hand, it is important to note that in line 4 of Algorithm 1, we aim to construct a column full-rank matrix $Q = \begin{bmatrix} \Lambda^{1/2} P^\top \\ \hat{0} \end{bmatrix}$, whereas the construction of $Q$ is also not unique since we can insert $d - m$ zero rows at any position within the original matrix $\Lambda^{1/2} P^\top$ to achieve the column full-rankness. The placement of these zero rows directly affects the result of feature selection.

Guided by Algorithm 1, we devise a more empirical approach by calculating $F$ with $F = \hat{F}(L^{-1})^\top$, which effectively tackles the above two concerns. By doing so, we avoid the non-uniqueness of eigendecomposition, thereby obtaining a solution that is as discrete as $\hat{F}$. Additionally, this approach ensures that the information of feature selection in $\hat{F}$ is retained within the column full-rank matrix.

Actually, $\boldsymbol{F}$ is an $\epsilon$-approximation of column-orthogonal matrix, since we have:

$$
\begin{aligned}
\boldsymbol{F}^\top \boldsymbol{F} &= \boldsymbol{L}^{-1} \hat{\boldsymbol{F}}^\top \hat{\boldsymbol{F}} (\boldsymbol{L}^{-1})^\top \\
&= \boldsymbol{L}^{-1} (\boldsymbol{A} - \epsilon \boldsymbol{I}_m)(\boldsymbol{L}^{-1})^\top \\
&= \boldsymbol{L}^{-1} \boldsymbol{A}(\boldsymbol{L}^{-1})^\top - \epsilon \boldsymbol{L}^{-1}(\boldsymbol{L}^{-1})^\top \\
&= \boldsymbol{L}^{-1} \boldsymbol{L} \boldsymbol{L}^\top (\boldsymbol{L}^{-1})^\top - \epsilon \boldsymbol{L}^{-1}(\boldsymbol{L}^{-1})^\top \\
&= \boldsymbol{I}_m - \epsilon \boldsymbol{L}^{-1}(\boldsymbol{L}^{-1})^\top.
\end{aligned}
\tag{S4}
$$

The experimental results in Section 4.2 verify the effectiveness of this approach in successfully avoiding duplicate feature selection.

## S3 Derivation of the Solution to Problem 5

Recall that we aim to solve the following problem to learn an adaptive $k$-NN graph:

$$
\begin{aligned}
\min_{\boldsymbol{S}} \operatorname{tr}(\hat{\boldsymbol{X}}^\top \boldsymbol{L}_S \hat{\boldsymbol{X}}) &+ \frac{\alpha}{2} \|\boldsymbol{S}\|_F^2, \quad \text{s.t. } \boldsymbol{S}\mathbf{1}_n = \mathbf{1}_n, s_{i,j} \geq 0, s_{i,i} = 0, \\
= \min_{s_{i,j}} \frac{1}{2} \sum_{i=1}^n \sum_{j=1}^n \|(\hat{\boldsymbol{x}}^i &- \hat{\boldsymbol{x}}^j)\|_2^2 s_{i,j} + \alpha_i s_{i,j}^2, \quad \text{s.t. } \sum_{j=1}^n s_{i,j} = 1, s_{i,j} \geq 0, s_{i,i} = 0.
\end{aligned}
\tag{S5}
$$

Based on $\hat{\boldsymbol{X}}$, we define the quantity $e_{i,j} = \|(\hat{\boldsymbol{x}}^i - \hat{\boldsymbol{x}}^j)\|_2^2$, then we solve each row in problem (S5) separately as:

$$
\begin{aligned}
\min_{s_{i,j}} \frac{1}{2} \sum_{j=1}^n e_{i,j} s_{i,j} &+ \alpha_i s_{i,j}^2 \quad \text{s.t. } \sum_{j=1}^n s_{i,j} = 1, s_{i,j} \geq 0, s_{i,i} = 0, \\
= \min_{s_{i,j}} \frac{1}{2} \sum_{j=1}^n (s_{i,j} &+ \frac{e_{i,j}}{2\alpha_i})^2 \quad \text{s.t. } \sum_{j=1}^n s_{i,j} = 1, s_{i,j} \geq 0, s_{i,i} = 0, \\
= \min_{\boldsymbol{s}^i} \frac{1}{2} \|\boldsymbol{s}^i &+ \frac{\boldsymbol{e}^i}{2\alpha_i}\|_2^2 \quad \text{s.t. } \boldsymbol{s}^i \mathbf{1}_n = 1, s_{i,j} \geq 0, s_{i,i} = 0.
\end{aligned}
\tag{S6}
$$

We first omit the constraint $s_{i,i} = 0$ and consider it later, and solve problem (S6) with the first two constraints, the Lagrangian function of which is as follows:

$$
\mathcal{L}(\boldsymbol{s}^i, \lambda_i, \boldsymbol{\beta}^i) = \frac{1}{2} \|\boldsymbol{s}^i + \frac{\boldsymbol{e}^i}{2\alpha_i}\|^2 - \lambda_i(\boldsymbol{s}^i \mathbf{1}_n - 1) - \sum_{j=1}^n s_{i,j} \beta_{i,j},
\tag{S7}
$$

where $\lambda_i$ and $\beta_{i,j}$ are Lagrange multipliers. The derivative of $\mathcal{L}(\boldsymbol{s}^i, \lambda_i, \boldsymbol{\beta}^i)$ w.r.t. $s_{i,j}$ is:

$$
\frac{\partial \mathcal{L}}{\partial s_{i,j}} = s_{i,j} + \frac{e_{i,j}}{2\alpha_i} - \lambda_i - \beta_{i,j}
\tag{S8}
$$

Then we have the Karush-Kuhn-Tucker(KKT) conditions [20] of problem (S7) as follows:

$$
\begin{cases}
s_{i,j} + \dfrac{e_{i,j}}{2\alpha_i} - \lambda_i - \beta_{i,j} = 0 \\
\displaystyle\sum_{j=1}^n s_{i,j} = 1 \\
s_{i,j} \geq 0 \\
\beta_{i,j} \geq 0 \\
\beta_{i,j} s_{i,j} = 0.
\end{cases}
\tag{S9}
$$

Then we have:

$$
s_{i,j} = (\lambda_i - \frac{e_{i,j}}{2\alpha_i})_+
\tag{S10}
$$

Recall that there are only $k$ nonzero elements in $s^i$ corresponding to the nearest neighbors of sample $i$, according to the constraint $\sum_{j=1}^n s_{i,j} = 1$ on $k$ nonzero entries in $s^i$, we have:

$$\sum_{j=1}^k (\lambda_i - \frac{e_{i,\sigma_j}}{2\alpha_i}) = 1 \Rightarrow \lambda_i = \frac{1}{k} + \frac{1}{k}\frac{e^i \delta_i^{(k)}}{2\alpha_i} \quad \text{with} \quad \delta_{i,j}^{(k)} = \text{Bool}(e_{i,j} \le e_{i,\sigma_k}), \qquad \text{(S11)}$$

where $\delta_i^{(k)}$ denotes the selection vector identifying the $k$ minimal values in $e^i$, and $\boldsymbol{\sigma} = [\sigma_1, \ldots, \sigma_n]$ denotes the sorting permutation over $e^i$, i.e. $e_{i,\sigma_1} \le \cdots \le e_{i,\sigma_n}$. Without loss of generality, we assume $e^i$ has no duplicates, namely $e_{i,\sigma_1} < \cdots < e_{i,\sigma_n}$. Considering the constraint $s_{i,i} = 0$, since $e_{i,i} = 0$ being the minimal value in $e^i$ holds for all samples, we replace $e_{i,i}$ with a sufficiently large value to skip over this trivial solution.

Substituting (S11) into (S10), we have:

$$s_{i,j} = (\frac{1}{k} + \frac{1}{k}\frac{e^i \delta_i^{(k)}}{2\alpha_i} - \frac{e_{i,j}}{2\alpha_i})_+ \qquad \text{(S12)}$$

Recall that there are only $k$ nonzero entries in $s^i$, we have

$$\frac{1}{k} + \frac{1}{k}\frac{e^i \delta_i^{(k)}}{2\alpha_i} - \frac{e_{i,\sigma_k}}{2\alpha_i} > 0, \quad \frac{1}{k} + \frac{1}{k}\frac{e^i \delta_i^{(k)}}{2\alpha_i} - \frac{e_{i,\sigma_{k+1}}}{2\alpha_i} \le 0. \qquad \text{(S13)}$$

Note that we assume $\alpha_i > 0$, then we have

$$\frac{1}{2}(ke^i \xi_i^{(k)} - e^i \delta_i^{(k)}) < \alpha_i \le \frac{1}{2}(ke^i \xi_i^{(k+1)} - e^i \delta_i^{(k)}) \quad \text{with} \quad \xi_{i,j}^{(k)} = \text{Bool}(e_{i,j} = e_{i,\sigma_k}), \quad \text{(S14)}$$

where $\xi_i^{(k)}$ is an indicator vector identifying the $k$-th minimal value in $e^i$. According to (S14), we set $\alpha_i$ as its maximal value as follows:

$$\alpha_i = \frac{1}{2}(ke^i \xi_i^{(k+1)} - e^i \delta_i^{(k)}). \qquad \text{(S15)}$$

Substituting S15 into Eq. S12, we have:

$$\begin{aligned}
s_{i,j} &= (\frac{1}{k} + \frac{1}{k}\frac{e^i \delta_i^{(k)}}{2\alpha_i} - \frac{e_{i,j}}{2\alpha_i})_+ \\
&= (\frac{2\alpha_i + e^i \delta_i^{(k)} - ke_{i,j}}{2k\alpha_i})_+ \\
&= (\frac{ke^i \xi_i^{(k+1)} - e^i \delta_i^{(k)} + e^i \delta_i^{(k)} - ke_{i,j}}{k(ke^i \xi_i^{(k+1)} - e^i \delta_i^{(k)})})_+ \\
&= (\frac{e^i \xi_i^{(k+1)} - e_{i,j}}{ke^i \xi_i^{(k+1)} - e^i \delta_i^{(k)}})_+
\end{aligned} \qquad \text{(S16)}$$

**Eq. (S16) is used for implementation in our code.** Note that since $e_{i,\sigma_1} < \cdots < e_{i,\sigma_k} < e_{i,\sigma_{k+1}} < \ldots e_{i,\sigma_n}$, we have

$$ke^i \xi_i^{(k+1)} - e^i \delta_i^{(k)} = ke_{i,\sigma_{k+1}} - \sum_{p=1}^k e_{i,\sigma_p} = \sum_{p=1}^k (e_{i,\sigma_{k+1}} - e_{i,\sigma_p}) > 0. \qquad \text{(S17)}$$

Then we obtain the solution of $s_{i,j}$ as

$$s_{i,\sigma_j} = \begin{cases} \dfrac{e^i \xi_i^{(k+1)} - e_{i,\sigma_j}}{ke^i \xi_i^{(k+1)} - e^i \delta_i^{(k)}}, & 1 \le j \le k \\ 0, & \text{otherwise} \end{cases}, \qquad \text{(S18)}$$

which is exactly the solution of Eq. (9) in our main paper:

$$s_{i,\sigma_j} = \frac{e^i \xi_i^{(k+1)} - e_{i,\sigma_j}}{ke^i \xi_i^{(k+1)} - e^i \delta_i^{(k)}} \cdot \text{Bool}(1 \le j \le k). \qquad \text{(S19)}$$

## S4 Connection to CLR

We note that a similar formulation to problem 6 has been proposed in [21] (coined CLR), which expects closer samples to have higher similarity. It aligns with the notion of "smoothness" as we mentioned in Section 2. However, our method differs from CLR in at least two crucial aspects: Firstly, CLR measures the distance quantity $e_{i,j}$ across all original features, making it more sensitive to the noise and irrelevant features in the original data. In contrast, our approach learns the graph structure using only informative features, resulting in enhanced robustness against noisy features. Secondly, it is important to note that CLR is proposed in the context of traditional machine learning, where optimization is straightforward, as $\boldsymbol{\delta}_i^{(k)}$ and $\boldsymbol{\xi}_i^{(k+1)}$ can be updated using off-the-shelf sorting algorithms. Different from CLR, problem 6 is introduced in the realm of deep learning, where conventional sorting algorithms are non-differentiable and not applicable. This poses a huge challenge in learning an adaptive $k$-NN graph in neural networks. To overcome this challenge, we proposed to transform the top-$k$ selection into a differentiable operator using the Optimal Transport technique.

## S5 Iterative Bregman Projections

In this paper, we employ the iterative Bregman projections [27] algorithm to solve the following problem:

$$\min_{\boldsymbol{\Gamma}} \langle \boldsymbol{C}, \boldsymbol{\Gamma} \rangle + \gamma \sum_{i,j} \Gamma_{i,j} \log \Gamma_{i,j}, \quad \text{s.t. } \boldsymbol{\Gamma}\mathbf{1}_{k+2} = \boldsymbol{\mu}, \boldsymbol{\Gamma}^\top \mathbf{1}_n = \boldsymbol{\nu}, \Gamma_{i,j} \geq 0. \tag{S20}$$

We first initialize two variables $\boldsymbol{u} \in \mathcal{R}^{k+2}$ and $\boldsymbol{K} \in \mathcal{R}^{n \times (k+2)}$ as $u_i = 1/(k+2)$ and $k_{i,j} = e^{-c_{i,j}/\gamma}$, respectively. Then based on the following formulations, we repeatedly updating $\boldsymbol{u}$ and $\boldsymbol{v}$ for $\zeta$ iterations:

$$\boldsymbol{v} = \frac{\boldsymbol{\mu}}{\boldsymbol{K}\boldsymbol{u}}, \quad \boldsymbol{u} = \frac{\boldsymbol{\nu}}{\boldsymbol{K}^\top \boldsymbol{v}}, \tag{S21}$$

where the division in Eq. (S21) is element-wise. In this paper, we set $\zeta$ as 200. After updating $\zeta$ iteration, we obtain the optimal transport plan $\boldsymbol{\Gamma}$ as

$$\boldsymbol{\Gamma} = \text{diag}(\boldsymbol{v})\boldsymbol{K}\text{diag}(\boldsymbol{u}). \tag{S22}$$

## S6 Supplementary Experimental Details

### S6.1 Implementation Details

All experiments are conducted on a server equipped with an RTX 3090 GPU and an Intel Xeon Gold 6240 (18C36T) @ 2.6GHz x 2 (36 cores in total) CPU.

#### S6.1.1 Synthetic Datasets

We generate three datasets for toy experiments: (1) Blobs, (2) Moons, and (3) Circles. For each dataset, we generate the first two features using the scikit-learn library [47] by adding noise sampled from $\mathcal{N}(0, 0.1)$. Additionally, we generate 18-dimensional noise features sampled from $\mathcal{N}(0, 1)$.

#### S6.1.2 Competing Methods

The implementation details of different methods, as well as their corresponding parameter selections are provided below:

- **Our Method**: Our method is implemented using the PyTorch framework [48]. We train our method using the Adam optimizer for 1000 epochs on all datasets, with the learning rate searched from $\{10^{-4}, 10^{-3}, 10^{-2}, 10^{-1}, 10^0, 10^1\}$. We search the parameter $\gamma$ in $\{10^{-3}, 10^{-2}, 10^{-1}\}$ and the parameter $k$ in $\{1, 2, 3, 4, 5\}$. Note that the implementation of differentiable top-$k$ selector is based on the code provided by [26] in `https://papers.nips.cc/paper_files/paper/2020/hash/ec24a54d62ce57ba93a531b460fa8d18-Abstract.html`, which provides a more memory-saving backward implementation compared to directly using the autograd method in PyTorch.

- Where to Pay Attention in Sparse Training (**WAST**) [41]: We use the official code released in `https://github.com/GhadaSokar/WAST`. The parameter settings were adopted in accordance with Appendix A.1 of the original paper. Specifically, we train each dataset for 10 epochs using stochastic gradient descent with a learning rate of 0.1 for all datasets except for SMK, where the learning rate is set to 0.01. For the parameter $\lambda$, we set $\lambda = 0.9$ on Madelon and PCMAC, $\lambda = 0.4$ on all image datasets, $\lambda = 0.1$ on all biological datasets except SMK, and $\lambda = 0.01$ on SMK. The remaining parameters are kept as they were in the original paper.

- Differentiable Unsupervised Feature Selection (**DUFS**) [14]: We use the official code released in `https://github.com/Ofirlin/DUFS` and use the parameter-free loss version of DUFS. For all datasets, we set $k = 2$, and train the method with SGD with a learning rate of 1 for 10000 epochs according to Appendix S7 in the original paper. We set the parameter $C = 5$ on all datasets except for SRBCT, COIL, and PIX10, where $C$ is set to 2.

- Concrete AutoEncoder (**CAE**) [8]: We use the official code released in `https://github.com/mfbalin/Concrete-Autoencoders`. Since we could not find too much description about the parameter settings on different datasets in the original paper, we run CAE with default settings in the code.

- AutoEncoder Feature Selector (**AEFS**) [9]: The original code provided by the authors is implemented in MATLAB, and it requires a prohibitively long time to run this method on MATLAB. Therefore, following the treatment in [41], we use the code provided by the authors of CAE in `https://github.com/Ofirlin/DUFS` (see experiments/generate_comparison_figures.py in their repository). We search the parameter $\alpha$ in $\{10^{-9}, 10^{-6}, 10^{-3}, 10^{0}, 10^{3}, 10^{6}, 10^{9}\}$, and the size of the hidden layer in $\{128, 256, 512, 1024\}$.

- Laplacian Score (**LS**) [13]: We use the official code released in `http://www.cad.zju.edu.cn/home/dengcai/Data/ReproduceExp.html#LaplacianScore`. We use the heat kernel for graph construction, and fix the size of neighbors $k$ as 5 for all datasets.

- Regularized Self-Representation (**RSR**) [42]: We use the official code released in `https://github.com/AISKYEYE-TJU/RSR-PR2015`. We search the parameter $\lambda$ in $\{10^{-9}, 10^{-6}, 10^{-3}, 10^{0}, 10^{3}, 10^{6}, 10^{9}\}$.

- Unsupervised Discriminative Feature Selection (**UDFS**) [43]: We use the code provided in `https://guijiejie.github.io/code.html`. We use the heat kernel for graph construction, and fix the size of neighbors $k$ as 5 for all datasets. We search the parameter $\gamma$ in $\{10^{-9}, 10^{-6}, 10^{-3}, 10^{0}, 10^{3}, 10^{6}, 10^{9}\}$.

### S6.1.3 Evaluation Workflow

The overall evaluation workflow in Section 4.3 is shown in Algorithm S2, which includes two steps:

1. Given the dataset $X$ and a prespecified feature number $m$, we first randomly split the dataset using an $8 : 2$ ratio and select features based on the training set $X_{tr}$ by FS method $\mathfrak{F}$ using different parameters $\Theta$, as shown in Algorithm S1. This allows us to obtain FS results under different parameter candidates $\theta_i$, along with the corresponding reduced training data $X'_{tr}$ and testing data $X'_{te}$. Based on the reduced data, we perform classification tasks on these datasets with the random forest, thereby obtaining the classification performance for each parameter combination. We select the parameter combination with the best classification performance as the optimal parameter $\theta^*$ for $\mathfrak{F}$.

2. Based on the optimal parameter $\theta^*$, we construct the FS model $\mathfrak{F}_{\theta^*}$ and evaluate its performance in different downstream tasks. To avoid randomness, we randomly split the dataset 10 times. With each random split, we use the training set $X_{tr}$ to select features, and obtain reduced training set $X'_{tr}$ and testing set $X'_{te}$. We use these sets for downstream tasks including classification, clustering, and reconstruction, and obtain corresponding performance metrics. For each downstream task, we calculate the average metric over 10 runs as the performance of $\mathfrak{F}$ for the given number $m$.

For each dataset, we vary the value of $m$ and follow the aforementioned procedure to obtain the corresponding performance. For each downstream task, we report the best metric and the corresponding feature number $m$ as the performance of the FS method in this downstream task.

---

**Algorithm S1** Param_tuning

---

**Input:** Training data $(\boldsymbol{X}_{tr}, \boldsymbol{y}_{tr})$, testing data $(\boldsymbol{X}_{te}, \boldsymbol{y}_{te})$, selected number $m$, FS method $\mathfrak{F}$, and parameter set $\boldsymbol{\Theta} = \{\theta_i\}$.
**Output:** Optimal parameter $\theta^*$.

1: **for** $\theta_i$ in $\boldsymbol{\Theta}$ **do**
2:      $\boldsymbol{\xi} = \mathfrak{F}_{\theta_i}(\boldsymbol{X}_{tr}, m);$           ▷ Determining selected features $\boldsymbol{\xi}$ by $\mathfrak{F}$ under the parameter $\theta_i$.
3:      $\boldsymbol{X}'_{tr} = \boldsymbol{X}_{tr}(:, \boldsymbol{\xi}), \boldsymbol{X}'_{te} = \boldsymbol{X}_{te}(:, \boldsymbol{\xi});$   ▷ Generating reduced datasets using selected features.
4:      ACC = RF($\boldsymbol{X}'_{tr}, \boldsymbol{y}_{tr}, \boldsymbol{X}'_{te}, \boldsymbol{y}_{te}$);▷ Evaluating selected features with random forest classifier.
5:      **if** ACC > ACC* **then**
6:          ACC* = ACC;
7:          $\theta^* = \theta_i;$
8:      **end if**
9: **end for**

---

---

**Algorithm S2** Overall evaluation workflow

---

**Input:** Original dataset $(\boldsymbol{X}, \boldsymbol{y})$, selected feature number $m$, FS method $\mathfrak{F}$, parameter set $\boldsymbol{\Theta} = \{\theta_i\}$, and downstream tasks $\mathfrak{T} = \{\mathcal{T}_i\}$.
**Output:**. Performance $\boldsymbol{M} = \{M_i\}$ in downstream tasks.

1: Partitioning the dataset into training data $(\boldsymbol{X}_{tr}, \boldsymbol{y}_{tr})$ and testing data $(\boldsymbol{X}_{te}, \boldsymbol{y}_{te})$.
2: Determining $\theta^*$ by Param_tuning($(\boldsymbol{X}_{tr}, \boldsymbol{y}_{tr}), (\boldsymbol{X}_{te}, \boldsymbol{y}_{te}), m, \mathfrak{F}, \boldsymbol{\Theta}$) in Algorithm S1;
3: **for** $j = 1 : 10$ **do**
4:      Partitioning the dataset into training data $(\boldsymbol{X}_{tr}, \boldsymbol{y}_{tr})$ and testing data $(\boldsymbol{X}_{te}, \boldsymbol{y}_{te})$.
5:      $\boldsymbol{\xi}^* = \mathfrak{F}_{\theta^*}(\boldsymbol{X}_{tr}, m);$
6:      $\boldsymbol{X}'_{tr} = \boldsymbol{X}_{tr}(:, \boldsymbol{\xi}^*), \boldsymbol{X}'_{te} = \boldsymbol{X}_{te}(:, \boldsymbol{\xi}^*);$
7:      **for** $\mathcal{T}_i$ in $\mathfrak{T}$ **do**
8:          $m\{i, j\} = \mathcal{T}_i(\boldsymbol{X}'_{tr}, \boldsymbol{y}_{tr}, \boldsymbol{X}'_{te}, \boldsymbol{y}_{te});$    ▷ Evaluating the performance in downstream tasks.
9:      **end for**
10: **end for**
11: **for** $\mathcal{T}_i$ in $\mathfrak{T}$ **do**
12:      $M_i = \text{Average}(m\{i, :\});$
13: **end for**

---

### S6.1.4    Evaluation Metrics

We employ two metrics in our experiments: the accuracy (ACC) and the root mean square error (RMSE) normalized by $d$.

The formulation of ACC is

$$\text{ACC}(\boldsymbol{y}, \hat{\boldsymbol{y}}) = \frac{\sum_{i=1}^{n} \text{Bool}(y_i = \hat{y}_i)}{n}, \tag{S23}$$

where $\boldsymbol{y} \in \mathcal{R}^n$ denotes the groundtruth labels and $\hat{\boldsymbol{y}} \in \mathcal{R}^n$ denotes the prediction label.

The formulation of RMSE normalized by $d$ is

$$\text{RMSE}(\boldsymbol{X}, \hat{\boldsymbol{X}}) = \sqrt{\frac{\sum_{i=1}^{n} \|\boldsymbol{x}^i - \hat{\boldsymbol{x}}^i\|_2^2}{n \times d}}, \tag{S24}$$

where $\boldsymbol{X} = [\boldsymbol{x}_1; \boldsymbol{x}_2; \ldots; \boldsymbol{x}_n]$ and $\hat{\boldsymbol{X}} = [\hat{\boldsymbol{x}}_1; \hat{\boldsymbol{x}}_2; \ldots; \hat{\boldsymbol{x}}_n]$ denote the original feature matrix and the reconstructed feature matrix, respectively.

### S6.1.5    Formulation of Heat Kernel Method

Here we describe the heat kernel (Heat) method compared in this paper. To implement Heat, we first compute the similarity matrix $\hat{\boldsymbol{S}}$ as follows:

$$\hat{s}_{i,j} = \exp(-\frac{\|\boldsymbol{x}^i - \boldsymbol{x}^j\|_2^2}{2\sigma^2}), \tag{S25}$$

Based on $\hat{S}$, we keep the $k$-nearest neighbors for each sample. Namely, for each sample $i$, we obtain its similarity vector $s^i$ as

$$s_{i,j} = \begin{cases} \hat{s}_{i,j}, & x^j \in \mathfrak{K}(x^i) \\ 0, & \text{otherwise} \end{cases}, \tag{S26}$$

where $\mathfrak{K}(x^i)$ denotes the $k$ nearest neighbors of $x^i$. When we need to calculate the Laplacian matrix using $S$ (for example, when we analyze the effect of DGL in Section 4.4), we use the symmetrized version of $S$:

$$\tilde{S} = \frac{S^\top + S}{2} \tag{S27}$$

## S6.2 Standard Deviations of Quantitative Analysis

Table S1, Table S2, and Table S3 exhibit the mean and the standard deviations of the results in Table 2 in Section 4.3.

Table S1: Classification results with standard deviations.

| Dataset | CAE | DUFS | WAST | AEFS | LS | RSR | UDFS | Our | AllFea |
|---|---|---|---|---|---|---|---|---|---|
| Madelon | 0.65±0.04 | 0.52±0.04 | 0.87±0.02 | 0.87±0.01 | 0.51±0.01 | 0.83±0.02 | 0.74±0.06 | 0.90±0.01 | 0.73±0.02 |
| PCMAC | 0.79±0.06 | 0.77±0.02 | 0.83±0.01 | 0.71±0.03 | 0.68±0.03 | 0.91±0.02 | 0.87±0.02 | 0.79±0.01 | 0.93±0.01 |
| COIL-20 | 1.00±0.01 | 1.00±0.00 | 1.00±0.00 | 1.00±0.00 | 0.98±0.01 | 1.00±0.00 | 1.00±0.00 | 1.00±0.00 | 1.00±0.00 |
| Yale | 0.72±0.05 | 0.72±0.05 | 0.70±0.05 | 0.70±0.06 | 0.67±0.07 | 0.72±0.08 | 0.71±0.06 | 0.81±0.05 | 0.75±0.06 |
| Jaffe | 0.97±0.02 | 0.97±0.01 | 0.97±0.01 | 0.97±0.01 | 0.97±0.03 | 0.98±0.02 | 0.98±0.02 | 1.00±0.01 | 0.98±0.02 |
| PIX10 | 0.99±0.02 | 0.98±0.03 | 0.99±0.02 | 0.98±0.03 | 0.97±0.06 | 0.97±0.04 | 0.97±0.04 | 1.00±0.00 | 0.97±0.03 |
| warpPIE10P | 0.96±0.04 | 0.96±0.03 | 0.95±0.03 | 0.98±0.04 | 0.96±0.05 | 0.98±0.03 | 0.97±0.05 | 0.99±0.02 | 0.98±0.04 |
| GLIOMA | 0.68±0.10 | 0.66±0.12 | 0.72±0.09 | 0.63±0.12 | 0.67±0.13 | 0.67±0.11 | 0.68±0.10 | 0.81±0.09 | 0.72±0.14 |
| LUNG | 0.86±0.05 | 0.87±0.05 | 0.87±0.04 | 0.87±0.05 | 0.92±0.04 | 0.93±0.05 | 0.91±0.06 | 0.94±0.03 | 0.90±0.05 |
| PROSTATE | 0.88±0.05 | 0.81±0.07 | 0.81±0.06 | 0.81±0.07 | 0.88±0.07 | 0.90±0.07 | 0.90±0.05 | 0.89±0.05 | 0.89±0.06 |
| SRBCT | 0.99±0.02 | 0.94±0.05 | 0.98±0.03 | 0.95±0.05 | 0.98±0.04 | 1.00±0.00 | 1.00±0.00 | 0.98±0.03 | 0.99±0.02 |
| SMK | 0.67±0.06 | 0.66±0.07 | 0.65±0.07 | 0.66±0.08 | 0.72±0.10 | 0.72±0.11 | 0.73±0.09 | 0.75±0.06 | 0.68±0.10 |

Table S2: Clustering results with standard deviations.

| Dataset | CAE | DUFS | WAST | AEFS | LS | RSR | UDFS | Our | AllFea |
|---|---|---|---|---|---|---|---|---|---|
| Madelon | 0.60±0.01 | 0.52±0.01 | 0.52±0.01 | 0.55±0.04 | 0.52±0.01 | 0.61±0.02 | 0.57±0.05 | 0.60±0.01 | 0.58±0.04 |
| PCMAC | 0.52±0.01 | 0.51±0.01 | 0.51±0.01 | 0.51±0.01 | 0.52±0.01 | 0.51±0.01 | 0.51±0.01 | 0.53±0.02 | 0.51±0.01 |
| COIL-20 | 0.69±0.03 | 0.59±0.04 | 0.65±0.03 | 0.60±0.04 | 0.53±0.04 | 0.59±0.03 | 0.60±0.04 | 0.66±0.03 | 0.63±0.05 |
| Yale | 0.55±0.06 | 0.55±0.07 | 0.56±0.06 | 0.54±0.06 | 0.58±0.06 | 0.60±0.08 | 0.58±0.05 | 0.62±0.04 | 0.61±0.08 |
| Jaffe | 0.85±0.07 | 0.80±0.06 | 0.83±0.05 | 0.80±0.07 | 0.76±0.09 | 0.83±0.08 | 0.83±0.09 | 0.87±0.06 | 0.82±0.06 |
| PIX10 | 0.86±0.05 | 0.79±0.04 | 0.85±0.06 | 0.79±0.04 | 0.85±0.10 | 0.72±0.08 | 0.81±0.09 | 0.87±0.07 | 0.78±0.05 |
| warpPIE10P | 0.55±0.05 | 0.42±0.03 | 0.44±0.04 | 0.53±0.06 | 0.55±0.05 | 0.57±0.06 | 0.49±0.08 | 0.51±0.05 | 0.45±0.05 |
| GLIOMA | 0.69±0.07 | 0.65±0.11 | 0.65±0.10 | 0.62±0.10 | 0.63±0.13 | 0.65±0.16 | 0.68±0.11 | 0.75±0.08 | 0.62±0.08 |
| LUNG | 0.64±0.07 | 0.64±0.11 | 0.62±0.10 | 0.66±0.06 | 0.65±0.11 | 0.69±0.10 | 0.61±0.05 | 0.72±0.12 | 0.69±0.11 |
| PROSTATE | 0.64±0.11 | 0.59±0.05 | 0.58±0.05 | 0.59±0.05 | 0.71±0.13 | 0.63±0.10 | 0.69±0.11 | 0.68±0.09 | 0.64±0.10 |
| SRBCT | 0.76±0.09 | 0.54±0.05 | 0.56±0.08 | 0.57±0.08 | 0.56±0.07 | 0.60±0.11 | 0.59±0.11 | 0.63±0.10 | 0.52±0.06 |
| SMK | 0.60±0.06 | 0.58±0.05 | 0.58±0.06 | 0.58±0.05 | 0.59±0.05 | 0.59±0.05 | 0.61±0.07 | 0.64±0.05 | 0.60±0.07 |

Table S3: Reconstruction results with standard deviations.

| Dataset | CAE | DUFS | WAST | AEFS | LS | RSR | UDFS | Our | AllFea |
|---|---|---|---|---|---|---|---|---|---|
| Madelon | 0.99±0.01 | 0.99±0.00 | 0.98±0.00 | 0.98±0.00 | 0.99±0.00 | 0.98±0.00 | 0.98±0.00 | 0.98±0.00 | 0.28±0.00 |
| PCMAC | 1.14±0.17 | 1.03±0.06 | 1.05±0.06 | 1.05±0.07 | 1.04±0.06 | 1.17±0.19 | 1.07±0.12 | 1.30±0.22 | 0.78±0.03 |
| COIL-20 | 0.48±0.03 | 0.41±0.02 | 0.43±0.01 | 0.41±0.01 | 0.48±0.02 | 0.45±0.02 | 0.41±0.02 | 0.38±0.02 | 0.27±0.01 |
| Yale | 0.63±0.03 | 0.56±0.02 | 0.56±0.02 | 0.56±0.02 | 0.78±0.04 | 0.57±0.03 | 0.60±0.03 | 0.54±0.02 | 0.50±0.01 |
| Jaffe | 0.31±0.05 | 0.25±0.05 | 0.26±0.05 | 0.25±0.05 | 0.33±0.08 | 0.26±0.05 | 0.25±0.05 | 0.22±0.01 | 0.23±0.04 |
| PIX10 | 0.49±0.03 | 0.43±0.04 | 0.46±0.05 | 0.43±0.02 | 0.63±0.05 | 0.46±0.04 | 0.47±0.05 | 0.39±0.03 | 1.20±0.23 |
| warpPIE10P | 0.30±0.03 | 0.26±0.02 | 0.27±0.02 | 0.26±0.02 | 0.45±0.06 | 0.27±0.03 | 0.26±0.02 | 0.25±0.01 | 0.25±0.04 |
| GLIOMA | 0.74±0.05 | 0.71±0.04 | 0.71±0.04 | 0.72±0.05 | 0.71±0.04 | 0.72±0.04 | 0.72±0.04 | 0.69±0.04 | 1.86±0.59 |
| LUNG | 0.94±0.06 | 0.77±0.02 | 0.77±0.02 | 0.78±0.02 | 0.82±0.04 | 0.81±0.02 | 0.79±0.02 | 0.80±0.03 | 0.82±0.28 |
| PROSTATE | 1.00±0.14 | 0.78±0.08 | 0.77±0.08 | 0.77±0.09 | 0.72±0.10 | 0.74±0.10 | 0.71±0.10 | 0.73±0.13 | 1.48±0.26 |
| SRBCT | 0.84±0.03 | 0.77±0.02 | 0.77±0.02 | 0.78±0.02 | 0.80±0.02 | 0.79±0.02 | 0.78±0.02 | 0.75±0.02 | 0.83±0.03 |
| SMK | 0.98±0.07 | 0.68±0.02 | 0.68±0.03 | 0.68±0.02 | 0.78±0.04 | 0.78±0.04 | 0.73±0.03 | 0.69±0.03 | 4.32±0.45 |

### S6.3 Reconstruction Results on PIX10

Fig. S1 presents the reconstruction results on PIX10 achieved by our method. We can see that our method is able to reconstruct the original images of 10000 dimensions reasonably well with only 300 features. Notably, the reconstructions capture important appearance details, including reflections, hair tips, and facial features, demonstrating the effectiveness of our method.

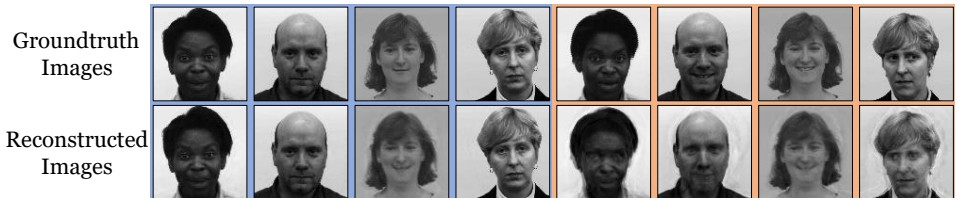

Figure S1: Reconstruction results on PIX10 with 300 features, where the training data are indicated in blue and testing data are indicated in light orange.

### S6.4 Graph Visualization

We visualize COIL-20 and Jaffe with t-SNE, and plot the graph structures obtained by different methods. The results are shown in Fig. S2, where red lines represent the intra-class connections and blue lines represent inter-class connections. Unlike "Heat" and "DGL only" that use the original features for visualization, we visualize the data points using only selected features. Remarkably, our method successfully achieves separable structures using t-SNE, demonstrating its ability to capture features that reflect the intrinsic data structure.

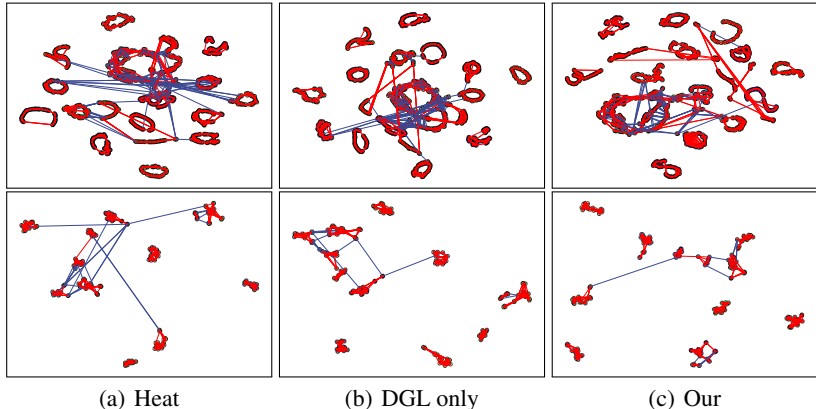

|         (a) Heat         |        (b) DGL only        |        (c) Our         |

Figure S2: Visualization with t-SNE, where the first row and the second row correspond to COIL-20 and Jaffe, respectively. Different classes are indicated with different colors. Intra-class connections are indicated in red and inter-class connections are indicated in blue. For each sample, we only present the connections to the 5 nearest neighbors.

### S6.5 Extended Results for Ablation Study

Table S4 and Table S5 present the clustering result and the reconstruction result of the ablation study on DGL in Section 4.4:

Table S4: Clustering results without DGL.

| Method | Madelon | PCMAC | Jaffe | PIX10 | GLIOMA | PROSTATE |
|--------|---------|-------|-------|-------|--------|----------|
| w/o DGL | 0.52±0.01 | 0.51±0.01 | 0.81±0.06 | 0.78±0.06 | 0.65±0.08 | 0.57±0.05 |
| Our | **0.60±0.01** | **0.53±0.02** | **0.87±0.06** | **0.87±0.07** | **0.75±0.08** | **0.68±0.09** |

Table S5: Reconstruction results without DGL.

| Method | Madelon | PCMAC | Jaffe | PIX10 | GLIOMA | PROSTATE |
|--------|---------|-------|-------|-------|--------|----------|
| w/o DGL | 0.99±0.00 | **1.07±0.10** | 0.26±0.05 | 0.45±0.02 | 0.74±0.03 | 0.79±0.08 |
| Our | **0.98±0.00** | 1.30±0.22 | **0.22±0.01** | **0.39±0.03** | **0.69±0.04** | **0.73±0.13** |

## S6.6 Additional Experiment: Parameter Sensitivity Analysis

In this section, we analyze the effect of the parameters of our method, including the learning rate, the number of nearest neighbors $k$, the hyperparameter $\gamma$ in the entropy regularized OT problem 14, and the selected feature number $m$. We use four real-world datasets, including a text dataset PCMAC, an artificial dataset Madelon, an image dataset Jaffe, and a biological dataset PROSTATE.

The analysis is based on the optimal parameter obtained in Section 4.3. For each dataset, we fix the values of the remaining parameters and vary the value of one parameter at a time. We retrain our method using the updated parameter combination and evaluate the corresponding FS result with the random forest. This allows us to observe the impact of different parameters on the performance of our method. For example, to analyze the effect of the learning rate on Madelon, we keep $k$, $\gamma$, and $m$ at their optimal values, then we vary the learning rate in $\{10^{-4}, 10^{-3}, 10^{-2}, 10^{-1}, 10^{0}, 10^{1}\}$ (the range as we described in Appendix S6.1.2), and evaluate their corresponding performance in the classification task with the random forest. The overall results are shown in Fig. S3, where the stars represent the results using optimal parameters.

One the one hand, we observe that the variations in the learning rate, $k$, $\gamma$ have little impact on the performance of our method across different datasets. This suggests that we can set a value within a proper range for these parameters, without the need to determine their values on different datasets. On the other hand, the most sensitive parameter is $m$, where a higher number of features contributes to better results, aligning with intuition and observations from existing literature. However, it is important to emphasize that more features are not always better. As demonstrated in Section 4.3, the FS methods consistently outperform AllFea in most classification and clustering tasks. Fewer features not only result in lower computational costs but also contribute to faster learning speeds. This suggests the need to adjust the value of $m$, for example, starting with a relatively small value and gradually increasing it until the model performance begins to decline.

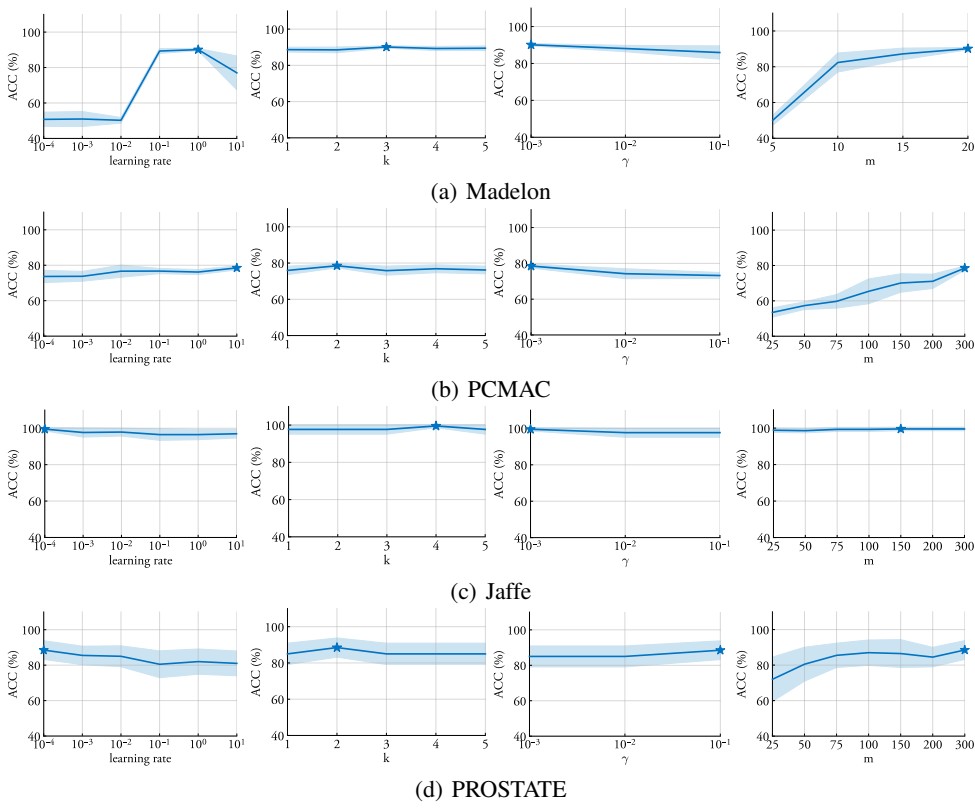

(a) Madelon

(b) PCMAC

(c) Jaffe

(d) PROSTATE

Figure S3: Parameter sensitivity analysis using the random forest, where the starred point denotes the performance on the optimal parameter combination.

