# OpenReview forum: "Joint Feature and Differentiable $ k $-NN Graph Learning using Dirichlet Energy"
_NeurIPS.cc/2023/Conference — NeurIPS 2023 poster_

### Official Review · Reviewer_MNAm · 2023-06-19

**Soundness:** 4 excellent
**Presentation:** 2 fair
**Contribution:** 4 excellent
**Rating:** 8
**Confidence:** 3

**Summary:**

The authors present a method to jointly learn important features as well as a $k$-nearest neighbour graph for data. They extensively motivate and evaluate their method.

**Strengths:**

- Great method, very novel and sensible.

- Excellent and thorough evaluation.

- Method is well-justified by theoretical explanations.

- Good ablation studies allow disentangling of the different components of the method.

- It is very admirable that the authors include the reconstruction metric in their main results, where their method does not perform as well. I agree with the authors that this is not a good metric for noisy data, but it is great that the authors do not try to hide their performance nonetheless.

**Weaknesses:**

- Very densely written, the paper should be more accessible than this. If the authors are able to improve the clarity of the writing, the impact of this work will greatly increase.

- To the point above, I would recommend a few sentences about the intuition behind the method.

**Overall**: Excellent work, some intuitive explanations about the method would greatly improve the reader's understanding of how the method works.

**Questions:**

- In equation 5, if we require $s_{i, i} = 0$, then why is $\mathbf{S} = I$ a solution?

**Limitations:**

The authors do a great job describing the limitations of their method.

---

> ### Author Rebuttal · Authors · 2023-08-08
>
> We greatly appreciate the reviewer's positive comments on our work. We provide our response below:
>
> ### Response to the point in Weaknesses
>
> **P1: "I would recommend a few sentences about the intuition behind the method."**
>
> **A1:** Thank you. The main focus of our work is to select features in neural networks using the Dirichlet Energy. The motivation behind our method is that noisy and irrelevant data will deteriorate the quality of the graph structure, therefore degrading the performance of feature selection. To address this issue, we propose a deep FS method that uses the Dirichlet energy to learn features and update the $k$-NN graph structure jointly, which is the first contribution of our work illustrated in the top panel of Fig. 2 in our paper. As we depicted in the constraints in problem (2), the result of feature selection should be discrete and unique, although traditional gumbel-softmax method is capable of approximating the discrete result, the issue of uniqueness is still not addressed. To this end, we propose the second contribution of our paper, namely, the algorithmically designed UFS module, as illustrated in the bottom left panel of Fig. 2. Moreover, we propose to learn the $k$-NN graph by minimizing the Dirichlet Energy. Note that the traditional $k$-NN learning methods cannot be employed in neural networks due to the nondifferentiability nature of sorting operation. To address this issue, we employ the OT technique to learn $k$-NN differently, which is the third contribution of the proposed method illustrated in the bottom right panel of Fig. 2. We have also listed the novelty of the proposed method in our response to Reviewer JXCq in Q1, kindly refer to this response for extra information. The proposed framework is not only a novel FS method, but also provides a new paradigm for differentiable graph learning, which is demanded by existing literature.
>
> ### Response to point in Questions
>
> **Q1: "In equation 5, if we require $s_{i,i}=0$, then why is $S=I$ a solution?"**
>
> **A1:** Thanks.  The trivial solution $S=I$ applies to $\\min_S tr(F^\\top LF)$. When the problem changed into
> $$
> \\min_S tr(F^\\top LF)\\ s.t. S1_n=1_n,s_{i,j}\\ge0,s_{i,i}=0,
> $$
> this problem is equivalent to solving each row independently as
> $$
> \\min_{s_{i,j}} \\frac{1}{2} \\sum_{j=1}^ne_{i,j}s_{i,j}\\ s.t. \\sum_{j=1}^ns_{i,j}=1,s_{i,j}\\ge0,s_{i,i}=0,
> $$
> which can be derived from Eq. (S4) and Eq. (S5) in Appendix S3. It is easy to obtain that only the nearest data point can serve as the neighbour of $\\boldsymbol x^i$ with probability $1$, while all the other data points will not be the neighbours of $\\boldsymbol x^i$, which is also not desirable in $k$-NN graph learning.

---

> > ### Comment · Reviewer_MNAm · 2023-08-14
> > **Thank you for your rebuttal**
> >
> > Thanks for your detailed rebuttal. I maintain my high rating and hope this work gets accepted. If the authors wish, I believe some of the explanations provided in the rebuttals should be included in the paper.

---

### Official Review · Reviewer_JXCq · 2023-06-29

**Soundness:** 2 fair
**Presentation:** 2 fair
**Contribution:** 2 fair
**Rating:** 4
**Confidence:** 3

**Summary:**

This paper proposes a deep FS method that simultaneously conducts feature selection and differentiable k-NN graph learning based on the Dirichlet Energy. The Dirichlet Energy identifies important features by measuring their smoothness on the graph structure, and facilitates the learning of a new graph that reflects the inherent structure in the new feature subspace during the training process using selected features. The authors employ the Gumbel Softmax technique and the Optimal Transport theory to address the non-differentiability issues of learning discrete FS results and learning k-NN graphs in neural networks, which theoretically makes our model applicable to other graph neural networks. Furthermore, the proposed framework is interpretable, since all modules are designed algorithmically.







**Strengths:**

The originality of this paper is satisfying, which proposes a deep FS method that simultaneously conducts feature selection and differentiable k-NN graph learning based on the Dirichlet Energy and the significance is mainly built on this.

The quality and clarity of this paper is good based on the clear presentation of the proposed method shown in Figure 2.

**Weaknesses:**

1. The biggest problem of this paper is the limited novelty in formulation of the proposed method, which is mainly obtained by combination of the existing works, i.e., dirichlet energy or the differentiable learner. The authors should better highlight their novelty in Section 3 of this paper, which make it significantly different from the existing works.

2. The compared methods in the experiment are not enough for comparison and the authors can add one or more methods for comparison to validate the effectiveness of the proposed method.

3. The adopted datasets in the experiment are almost with small scales. The largest number of the adopted datasets is 2600, i.e., madelon. The authors can add one or more datasets with large scales for comparison in the experiment.

4. The experimental improvements of the proposed method compared with the existing methods are not obvious, i.e, the proposed method on SRBCT is 0.98 while UDFS is 1.00.

5. The authors can further explain the interpretability of the proposed method in the corresponding section , which is also an advantage of the given network.

**Questions:**

1. The biggest problem of this paper is the limited novelty in formulation of the proposed method, which is mainly obtained by combination of the existing works, i.e., dirichlet energy or the differentiable learner. The authors should better highlight their novelty in Section 3 of this paper, which make it significantly different from the existing works.

2. The compared methods in the experiment are not enough for comparison and the authors can add one or more methods for comparison to validate the effectiveness of the proposed method.

3. The adopted datasets in the experiment are almost with small scales. The largest number of the adopted datasets is 2600, i.e., madelon. The authors can add one or more datasets with large scales for comparison in the experiment.

4. The experimental improvements of the proposed method compared with the existing methods are not obvious, i.e, the proposed method on SRBCT is 0.98 while UDFS is 1.00.

5. The authors can further explain the interpretability of the proposed method in the corresponding section , which is also an advantage of the given network.

**Limitations:**

1. The biggest problem of this paper is the limited novelty in formulation of the proposed method, which is mainly obtained by combination of the existing works, i.e., dirichlet energy or the differentiable learner. The authors should better highlight their novelty in Section 3 of this paper, which make it significantly different from the existing works.

2. The compared methods in the experiment are not enough for comparison and the authors can add one or more methods for comparison to validate the effectiveness of the proposed method.

3. The adopted datasets in the experiment are almost with small scales. The largest number of the adopted datasets is 2600, i.e., madelon. The authors can add one or more datasets with large scales for comparison in the experiment.

4. The experimental improvements of the proposed method compared with the existing methods are not obvious, i.e, the proposed method on SRBCT is 0.98 while UDFS is 1.00.

5. The authors can further explain the interpretability of the proposed method in the corresponding section , which is also an advantage of the given network.

---

> ### Author Rebuttal · Authors · 2023-08-08
>
> We would like to thank the Reviewer for the feedback and suggestions.  Below we answer all questions and provide some additional experimental results.
>
>
>
> **Q1: Concern about the limited novelty of the proposed method**
>
> **A1:**
>
> Thank you for the suggestion about highlighting the novelty in Section 3. However, we respectfully disagree that our method has limited novelty. Our novelty lies in at least three aspects.
>   - Firstly, the proposed method is a model-driven framework, the design of which are motivated naturally by the issue of noisy and irrelevant data. It is noteworthy that each module in our framework is related to a specific algorithm design, kindly refer to our response to Q5.
>   - Secondly, the proposed UFS module is novel. Note that Equation (4) forms the core idea of CAE [1], a notable baseline advocating the use of Gumbel-softmax for feature selection (FS). However, CAE was criticized for duplicate selection [2, Sec. 2.2], as far as we know, few studies have addressed this issue. While UFS can proposes an improvement to CAE, with its efficacy demonstrated in our toy experiment.
>   - Thirdly, the proposed differentiable $k$-NN graph learning is novel. While OT techniques have been used for sorting before, to our knowledge, they have not been used for $k$-NN graph learning. In fact, there are few studies on the differentiable $k$-NN graph, and this point is supported by literature [3]. In order to obtain a differentiable graph, the authors in [3] simply relax the hard connection $w_{u,v}\\in\\{0,1\\}$ between node $u$ and node $v$ into a soft connection $w_{u,v}\\in(0,1)$, which potentially create a dense graph. In contrast, our method can generate a sparser graph structure, which provides a new paradigm for differentiable graph learning.
>
> We're gratified by reviewers EQBi and MNAm acknowledging the novelty of our approach, while also recognizing MNAm's critique regarding the density of our writing. This may have obscured full comprehension of our contributions. We will add more explainations to make our method more accessible in the revised version.
>
> **Ref:**
>
> [1] Balın, M. F., et al. (2019, May). Concrete autoencoders: Differentiable feature selection and reconstruction. in Proc. ICML (pp. 444-453).
>
> [2] Atashgahi, Z., et al. (2022). Quick and robust feature selection: the strength of energy-efficient sparse training for autoencoders. Machine Learning, 1-38.
>
> [3] Miao, S., et al. (2022). Interpretable geometric deep learning via learnable randomness injection. in Proc. ICLR, 2023.
>
>
>
> **Q2: Concern about the compared methods**
>
> **A2:** Thank you for the comment. We have added the results of two popular FS methods (FIRDL [1] and STG [2]) on twelve datasets. The complete result is provided in Table 1 of the attached PDF file, below we provide some statistical results to show the superior performance of our method:
>
> | Task           | Methods      | STG  | FIRDL | Our     |
> | -------------- | ------------ | ---- | ----- | ------- |
> | Classification | Average rank | 2.1  | 2.3   | **1.1** |
> |                | Top-1 counts | 2    | 1     | **11**  |
> | Clustering     | Average rank | 2.3  | 2.3   | **1.1** |
> |                | Top-1 counts | 2    | 1     | **11**  |
> | Reconstruction | Average rank | 1.7  | 1.8   | **1.8** |
> |                | Top-1 counts | 5    | 5     | **7**   |
>
> **Ref:**
>
> [1] Wojtas, M., et al. (2020). Feature importance ranking for deep learning. in Proc. NIPS, 33, 5105-5114.
>
> [2] Yamada, Y., et al. (2020, November). Feature selection using stochastic gates. in Proc. ICML (pp. 10648-10659).
>
> **Q3: Concern about the datasets**
>
> **A3:** Thanks. As we mentioned in Discussion in our paper, the major limitation of our method is the lack of scalability, for which we did not adopt large datasets. We will work on this issue in our future study.
>
> **Q4: Concern about the experimental improvements of the proposed method**
>
> **A4:** Thank you. Given that Feature Selection (FS) outputs are typically used for various downstream tasks, and the specific task is often unknown in practical applications, the goal of FS is usually to select features that perform well across multiple tasks, which is named the 'unsupervised nature' of FS by existing literature [1].  Given this, we believe that evaluating the overall performance across various tasks and datasets provides a more comprehensive demonstration of a method's efficacy. Hence, our approach exhibits superior performance relative to existing methods (evident from the Average Ranking and # Top 1 in Table 2).
>
> **Ref:**
>
> [1] Balın, M. F., et al. (2019, May). Concrete autoencoders: Differentiable feature selection and reconstruction. in Proc. ICML (pp. 444-453).
>
> **Q5: Concern about the interpretability of the proposed method**
>
> **A5:** Thank you. Our work is rooted in model-driven neural networks, related work can be seen in [1] [2]. Interpretability here signifies comprehending the function of each module during learning.  Unlike most deep learning networks with complex components that are that are tough to decipher, each core module in our framework has an algorithmic, physically meaningful design, which greatly facilitates observing and understanding the network's internal operations during the learning process.
>
> Specifically, UFS, designed from Algorithm 1 and Proposition 3.1, produces an output resembling an orthogonal and discrete matrix (as detailed in our response to Reviewer f5Nx in P3). Moreover, the design of DGL, based on problem (5), assures that the learned graph reflects the intrinsic structure of the selected features. We will add more explanations about interpretability in the revised version.
>
> **Ref:**
>
> [1] Xie, Q., et al. (2020). MHF-Net: An interpretable deep network for multispectral and hyperspectral image fusion. IEEE Trans. Pattern Anal. Mach. Intell., 44(3), 1457-1473.
>
> [2] Wang, H., et al. (2020). A model-driven deep neural network for single image rain removal. in Proc. CVPR (pp. 3103-3112).

---

> > ### Comment · Reviewer_JXCq · 2023-08-17
> >
> > I appreciate the authors for their replies (i.e., performing experiments for validation by adding two recent works). The replies regarding the novelty in formulation still does not fully convince me and I keep my rating.

---

### Official Review · Reviewer_f5Nx · 2023-07-06

**Soundness:** 2 fair
**Presentation:** 3 good
**Contribution:** 2 fair
**Rating:** 5
**Confidence:** 4

**Summary:**

This article proposes an unsupervised feature selection method by minimizing the Dirichlet energy, and the energy function is on the other hand based on the k-NN graph computed from the selected features. In this sense, the features and the k-NN graph are jointly learned. To avoid discrete operations, the author(s) use Gumbel softmax to approximate the selection operation, and use OT-based sorting to construct k-NN graph. The proposed method is validated using a series of numerical experiments.

**Strengths:**

Overall the proposed method is an interesting joint feature selection and graph learning method, and the whole procedure is differentiable, thus useful for downstream analyses.

**Weaknesses:**

1. One major weakness of the proposed algorithm is its large computational cost, which the author(s) have mentioned in the article. The bottleneck seems to be in the sorting part. In fact, there are different variants of differentiable sorting, and it has been shown in some reports that OT-based sorting can be slow [1]. The author(s) may try other differentiable sorting algorithms such as [1], which has $O(n\log n)$ forward complexity and $O(n)$ backward complexity.
2. There exist several inconsistencies between the proposed method and the actual implementation, such as the symmetrized k-NN graph (page 2) and the actual computation of the $F$ matrix in page 5. Although the author(s) have explained their motivation, it downgrades the rigor of the method. Also Proposition 3.1 looks irrelevant if the actual $F$ is not computed in this way.
3. Following point 2, I think a deeper reason for such inconsistencies is that the optimization problem (2) is difficult by nature, and Section 3.1.1 and Section 3.1.2 are tackling the constraints in (2) separately, not jointly. For example, Section 3.1.1 attempts to make $F$ approximately discrete, and Section 3.1.2 is concerned with $F^{T}F=I$. It is very likely that they are not met simultaneously using the current algorithm.

[1] Blondel, M., Teboul, O., Berthet, Q., & Djolonga, J. (2020). Fast differentiable sorting and ranking. In International Conference on Machine Learning (pp. 950-959).

**Questions:**

1. I think both the title of Section 3.1.1 and the statement "having selected *exact* original features" in Section 3.1.2 need to be modified, as the Gumbel softmax technique is an approximation to the discrete optimization problem, not necessarily an exact one.
2. The author(s) may consider a direct convex relaxation of the constraints in (2), which at least jointly handles the two different constraints.
3. Some comments on the points in the weaknesses section would be helpful.

**Limitations:**

The author(s) have remarked that the proposed method may be difficult to scale up.

---

> ### Author Rebuttal · Authors · 2023-08-08
>
> We thank the reviewer for the constructive feedback and positive comments. Below we provide our response:
>
> ### Response to the points in Weaknesses
>
> **P1: Suggestion about a more computationally efficient  method for differentiable sorting in Weakness 1**
>
> **A1:** Thank you for this constructive comment. Reference [1] proposed to construct differentiable sorting operators as projections onto the permutahedron, and the $\\mathcal{O}(n \\log n)$ time and $\\mathcal{O}(n)$ space complexity was achieved with isotonic optimization. We will try this method in our future work. However, recall that there are $n$ sorting tasks in our network since we have $n$ samples, a space complexity of at least $\\mathcal{O}(n^2)$ would be required to store intermediate variables. While graph neural networks mitigate memory overhead by subgraph sampling [2], the neighbours of each sample in our method are determined on global information, which requires full information being loaded at each iteration. Hence, we have mentioned batch learning as a future research direction in our Discussion section for its potential to enhance scalability.
>
> **Ref:**
>
> [1] Blondel, M., et al. (2020). Fast differentiable sorting and ranking. in Proc. ICML (pp. 950-959).
>
> [2] Hamilton, W., et al. (2017). Inductive representation learning on large graphs. in Proc. NIPS, 30.
>
> **P2: Concern about the symmetrized k-NN graph in Weakness 2**
>
> **A2:** Thank you. The use of $\\hat{S} = (S+S^\\top)/2$ is indeed a common practice in Spectral graph methods, kindly see [1, Sec. 2.2] and [2, Sec. 2.1] for example. The underlying rationale is that the $k$-NN graph we learned is a directed graph, whereas the computation of the Laplacian matrix is based on the undirected graph. To make the graph undirected, a common approach is to ignore the directions of the edges. That is, we connect $x^i$ and $x^j$ if $x^i$ is among the $k$-NN of $x^j$ or $x^j$ is among the $k$-NN of $x^i$. After symmetrization, the Dirichlet Energy will select features based on the symmetrized undirected graph $\\hat{S}$, instead of the original graph $S$. We will clarify this in the revised version.
>
> **Ref:**
>
> [1] Von Luxburg, U. (2007). A tutorial on spectral clustering. Stat. Comput., 17, 395-416.
>
> [2] He, X., et al. (2005). Laplacian score for feature selection. in Proc. NIPS, 18.
>
> **P3: Concern about the actual computation of F and the Proposition 3.1  in Weakness 2**
>
> **A3:** Thank you, we agree that the actual computation of $F$ downgrades the rigor of the method, but we respectfully disagree that Proposition 3.1 is irrelevant to the actual computation. The significance of Proposition 3.1 is twofold.
>
> Firstly, since Cholesky decomposition is conditional, Eq. (S1) in Proposition 3.1 confirms that we are able to obtain a lower triangle matrix $L$ for any real matrix $\\hat{F}$, as $\\epsilon>0$ guarantees the positive-definiteness of $F^\\top F+\\epsilon{I}_m$.
>
> Secondly, Proposition 3.1 validates the efficacy of both Algorithm 1 and the actual computation in solving the orthogonal constraint. Actually, $F = \\hat{F}(L^{-1})^\\top$ is an approximation of Algorithm 1 since we have
>
> $${F}^\\top{F} =L^{-1}\\hat{F}^\\top\\hat{F}(L^{-1})^\\top=L^{-1}(A-\\epsilon{I}_m)(L^{-1})^\\top=L^{-1}A(L^{-1})^\\top-\\epsilon L^{-1}(L^{-1})^\\top=L^{-1}LL^\\top(L^{-1})^\\top-\\epsilon L^{-1}(L^{-1})^\\top=I_m-\\epsilon L^{-1}(L^{-1})^\\top,$$
>
> which can be viewed as an $\\epsilon$-approximation to orthogonality. While it cannot strictly attain orthogonality, the toy experiment nonetheless indicates that this design is effective for unique feature selection.
>
> ### Response to the points in Questions
>
> **Q1: "... both the title of Section 3.1.1 and the statement "having selected *exact* original features" in Section 3.1.2 need to be modified,..."**
>
> **A1:** Thank you for the comment. We will modify the title and the statement in the revised version.
>
> **Q2: "The author(s) may consider a direct convex relaxation of the constraints in (2), ..."**
>
> **A2:** Thank you for this valuable comment. Inspired by reference [1], below we provide a relaxed version of the constraint in problem (2) in our paper. Instead of learning the selection matrix $F$, we learn a discrete vector $\mathbf v$ containing only $k$ non-zero components:
> $$
> \\min_{{\mathbf v}} \\mathrm{tr}(\\hat{{X}}^\\top{L}\\hat{{X}})\\quad\\mathrm{s.t.}\\  \\hat{{X}} = {X}\\mathrm diag({\mathbf v}), {1}_d^\\top{\mathbf v}=k,{\mathbf v}\\in\\{0,1\\}^d
> $$
> Based on this, we can relax the discrete constraint to be the intersection of a solid cube and a shifted $\\ell_2$-sphere:
> $$
> {\mathbf v}\\in\\{0,1\\}^d\\Leftrightarrow\\{{\mathbf v}|{\mathbf v}\\in[0,1]^d\\}\\cap\\{{\mathbf v}|\\|{\mathbf v}-({1}_d/2)\\|^2_2=d/4\\},
> $$
> as illustrated in Fig. 1 of the attached pdf file. While [1] suggests the relaxed problem is readily solved using the ADMM method, it's crucial to highlight their optimization is for traditional ML models. To implement the alternating optimization process into a neural network, we can design modules similar to ADMM-net [2], where each module represents an optimization step and is iteratively repeated, kindly see [2] [3] for example.
>
> **Ref:**
>
> [1] Zhang, X., et al. (2020). Top-k feature selection framework using robust 0–1 integer programming. IEEE Trans. Neural Netw. Learn. Syst., 32(7), 3005-3019.
>
> [2] Sun, J., et al. (2016). Deep ADMM-Net for compressive sensing MRI. in Proc. NIPS, 29.
>
> [3] Lin, Z., wt al. (2011). Linearized alternating direction method with adaptive penalty for low-rank representation. in Proc. NIPS, 24.
>
> ### Extra words
> We are grateful for these insightful comments. Despite limited space leading to a dense presentation, as noted by reviewer MNAm, we have endeavored to clearly detail our approach. We believe that this transparency can spur the evolution of improved solutions, just like P1 and Q2. Such insights are invaluable,  as they significantly contribute to the continuous refinement of our work.

---

> > ### Comment · Reviewer_f5Nx · 2023-08-17
> >
> > I would like to thank the author(s) for their careful response and detailed clarification. The rebuttal as well as other reviewers' comments gives a clearer picture of the proposed method, and I expect that this framework has great potential for end-to-end feature selection in deep learning. On the other hand, there are indeed some limitations of the current work as pointed out in the reviews, including the scalability of the method, formulation of the constraints, reduced rigor for practical implementation, etc. Judging from both sides, I would like to keep my previous rating as borderline accept.
> >
> > I think one possible direction to enlarge the impact of this work is to design some approximate but more efficient algorithms for better scalability, for example, a differentiable approximate k-NN algorithm. Feature selection itself is a hard combinatorial optimization problem, and I agree that at least some sort of approximation needs to be used. But after paying the price of reduced rigor, there should be some visible benefits on other aspects. For this framework, I believe the computational cost is a key factor to consider.

---

### Official Review · Reviewer_EQBi · 2023-07-07

**Soundness:** 4 excellent
**Presentation:** 4 excellent
**Contribution:** 3 good
**Rating:** 7
**Confidence:** 3

**Summary:**

In this paper, the authors propose an unsupervised feature selection method using Dirichlet energy. The proposed method learns the KNN graph and the feature selection jointly to reduce the influence on the feature selection quality by the noisy and irrelevant features.

The feature selection component minimizes the Dirichlet energy with Gumbel Softmax smoothed one-hot feature selection matrix. To ensure each resulting feature dimension is unique, the column orthogonality is enforced by linear decompositions.

The graph learning component also minimizes the Dirichlet energy. To avoid a trivial solution (graph with only self-connecting edges), Tikhonov regularization is applied to ensure the probabilistic behavior of similarity metrics and no self-loops.

Empirical results on both synthetic datasets and real-world applications show that the proposed method can effectively select useful features and has better performance than other baseline methods.

**Strengths:**

* The paper is clearly motivated and well-structured. The derivations are clear and in detail. Overall the paper presentation is good.

* The proposed method is interesting and novel, and the effectiveness of the proposed method is supported by both synthetic and real-world datasets.

* The authors also discussed the limitations of the method and lies out the future directions to improve.

**Weaknesses:**

* As the authors mentioned in the discussion section, the method does not scale to large-scale problems.

* The method is under the assumption that the features are continuous and of low dimension, whereas many real-world problems may face noncontinuous (e.x. categorical) or high dimensional (e.x. sparse BOW) features.

**Questions:**

* How can the method be generalized to handle noncontinuous categorical features?

* Is a constant feature a trivial minimizer for the Dirichlet energy? How does the proposed method handle this?

**Limitations:**

See the weaknesses above.

---

> ### Author Rebuttal · Authors · 2023-08-08
>
> We would like to thank the reviewer for the positive comments.  We provide our response below.
>
> ### Response to points in Questions
>
> **Q1: "How can the method be generalized to handle noncontinuous categorical features?"**
>
> **A1:** Thanks. Recall that Dirichlet Energy is inherently reliant on the manifold distribution of continuous features. Contrastingly, discrete data does not manifest a discernible distribution characteristic. The most straightforward example is within categorical features, where the distance between type 1 and type 3 is not inherently larger than that between type 1 and type 2. Therefore, we cannot directly apply continuous measurements for categorical features.
>
> In order to incorporate discrete data into Dirichlet Energy calculations, we can first map the discrete data onto a continuous vector space, and then measure the distances between different vectors within this space. Specifically, suppose we have a feature set $\\mathbb{x}=\\mathbb{x}\_{con}\\cup\\mathbb{x}\_{cate}$ composed of continuous features $\\mathbb{x}\_{con}$ and categorical features $\\mathbb{x}\_{cate}$. For the categorical feature $\\hat{\\boldsymbol x}\_p$, the Dirichlet Energy in Eq. (1) can be extended into the following form:
> $$
> \\mathcal{L}\_{dir}(\\hat{\\boldsymbol x}\_p) = \\frac{1}{2}\\sum\_{i=1}^n\\sum\_{j=1}^ns\_{i,j}\\|\\mathfrak{f}(\\hat{x}\_{i,p})-\\mathfrak{f}(\\hat{x}\_{j,p})\\|\_2^2
> $$
> where $\\mathfrak{f}$ is a nonlinear map function. For example, we can first convert the categorical feature $\\hat{x}\_{i,p}$ into a one-hot vector, and then further map the one-hot vector into a continuous $h$-dim vector $\\boldsymbol z\_{(i,p)}$ with a MLP, namely, $\\boldsymbol z\_{(i,p)} = \\mathfrak{f}(\\hat{x}\_{i,p})$. Consequently, the distance between $\\hat{x}\_{i,p}$ and $\\hat{x}\_{i,q}$ can be measured by the $\\ell\_{2}$-norm distance between $\\boldsymbol z\_{(i,p)}$ and $\\boldsymbol z\_{(j,p)}$. Accordingly, the calculation of the distance matrix in line 163 in our paper is changed into
> $$
> e\_{i,j}=\\sum\_{\\hat{\\boldsymbol x}\_p\\in \\mathbb{x}\_{cate}}\\|\\mathfrak{f}(\\hat{x}\_{i,p})-\\mathfrak{f}(\\hat{x}\_{j,p})\\|\_2^2+\\sum\_{\\hat{\\boldsymbol x}\_q\\in \\mathbb{x}\_{con}}(\\hat{x}\_{i,q}-\\hat{x}\_{j,q})^2.
> $$
>
> **Q2: "Is a constant feature a trivial minimizer for the Dirichlet energy? How does the proposed method handle this?"**
>
> **A2:** Yes, according to Eq. (1) in our paper, the constant feature is a trivial solution to the Dirichlet Energy. However, as stated in the first paragraph in Section 2 (line 69), we assume that all features have zero means and normalized variances, namely, $\\boldsymbol 1\_n^\\top\\boldsymbol x\_p=0$ and $\\boldsymbol x\_p^\\top\\boldsymbol x\_p=1$. This practice is not unusual and is frequently employed during feature preprocessing in various Feature Selection studies, e.g., see [1], [2], and [3].  Under this assumption, constant features, which cannot satisfy this condition, will be dismissed at the preprocessing stage. We will clarify this in the revised version.
>
> **Ref:**
>
> [1] Lindenbaum, O., et al. (2021). Differentiable unsupervised feature selection based on a gated laplacian. in Proc. NIPS, 34, 1530-1542.
>
> [2] Sokar, G., et al. (2022). Where to pay attention in sparse training for feature selection?. in Proc. NIPS, 35, 1627-1642.
>
> [3] Nie, F., et al. (2010). Efficient and robust feature selection via joint ℓ2, 1-norms minimization. in Proc. NIPS, 23.

---

> > ### Comment · Reviewer_EQBi · 2023-08-21
> >
> > I'd like to thank the authors for their detailed reply and discussion. I will keep my score and vote for accept.

---

### Author Rebuttal · Authors · 2023-08-09

We thank all the reviewers for the positive reviews and constructive comments that help us to emphasize the contributions of our approach. We are encouraged to hear that the reviewers found the approach  **interesting** (Reviewers EQBi, f5Nx), **novel** (Reviewers EQBi, MNAm), and **well-motivated** (Reviewers EQBi, MNAm). We appreciate that the reviewers acknowledge that the paper is **well-presented** (Reviewers EQBi, JXCq). In response to the thoughtful comments, we have addressed them one by one in the individual responses.

In particular, we have highlighted our contribution in our response to Reviewer JXCq in Q1 and also described the intuition of the proposed method in the response to Reviewer MNAm in P1, please see these responses for detailed information.

**Below we have uploaded a pdf file that includes the illustration of the relaxed constraint in our response to Reviewer f5Nx in Q2, as well as the detailed experimental results in our response to Reviewer JXCq in Q2.**

Thanks again for all of your valuable suggestions, we appreciate the reviewers' time to check our response.

---

### Comment · Area_Chair_GsAH · 2023-08-11
**Discussion period**

Dear reviewers and authors,

Thank you very much for your work on this submission and its evaluation. Now that the authors have responded to the reviews, I *strongly encourage* the reviewers to acknowledge the review, to look at other reviews and rebuttals for this submission, and to adjust their scores if needed. Thanks to those that have already done so.

Authors have the possibility to reply if further questions are needed, until the 16th.

Thank you very much to all,

Area Chair

---

### Decision · Program_Chairs · 2023-09-21

**Decision:**

Accept (poster)

**Comment:**

There was some disagreement between reviewers on this submission, which led me to review it in depth myself, and while I agree that it has some limitations, I still think that it should be accepted **on the condition** that it addresses more clearly its limitations, mainly

- Issues with scale, and why a batch approach is not considered, and how it could be

- More in-depth literature review of differentiable discrete operators, as mentioned by the reviewers, including

[1] Blondel, M., Teboul, O., Berthet, Q., & Djolonga, J. (2020). Fast differentiable sorting and ranking.

[2] Q Berthet, M Blondel, O Teboul, M Cuturi, JP Vert, F Bach (2020) Learning with differentiable perturbed optimizers

[3] M Vlastelica, A Paulus, V Musil, G Martius, M Rolinek (2020) Differentiation of blackbox combinatorial solvers

[4] M Paulus, D Choi, D Tarlow, A Krause, CJ Maddison (2020) Gradient estimation with stochastic softmax tricks